



# Choosing an Optimal $\beta$ Factor for Relaxed Eddy Accumulation Applications Across Vegetated and non-Vegetated Surfaces

Teresa Vogl[1,2], Amy Hrdina[3], and Christoph K. Thomas[2]

[1]University of Leipzig, Institute for Meteorology, 04103 Leipzig, Germany
[2]University of Bayreuth, Micrometeorology Group, 95440 Bayreuth, Germany
[3]Department of Civil and Environmental Engineering, Massachusetts Institute of Technology, Cambridge, MA, USA

**Correspondence:** Teresa Vogl (teresa.vogl@uni-leipzig.de)

**Abstract.** Accurately measuring the turbulent transport of reactive and conservative greenhouse gases, heat, and organic compounds between the surface and the atmosphere is critical for understanding trace gas exchange and its response to changes in climate and anthropogenic activities. The Relaxed Eddy Accumulation (REA) method enables measuring the land surface exchange when fast-response sensors are not available, broadening the suite of trace gases that can be investigated. This study

evaluates a variety of different REA approaches with the goal of formulating universally applicable recommendations for an optimal choice of the $\beta$ factor in combination with a suitable deadband. The $\beta$ factor scales the concentration differences to the flux, and its choice is central to successfully using REA. Deadbands are used to select only certain turbulent motions to compute the flux. Observations were collected across three contrasting ecosystems offering stark differences in scalar transport and dynamics: A mid-latitude grassland ecosystem in Europe, a loose gravel surface of the Dry Valleys of Antarctica, and

a spruce forest site in the European mid-range mountains. We tested a total of three different REA models for the $\beta$ factor: The first method derives $\beta_0$ based on a proxy for which high-frequency observations are available (sensible heat). The second method employs the approach of Baker et al. (1992), which computes $\beta_w$ solely based upon the vertical wind statistics. The third method uses a constant $\beta$ derived from long-term averaging of the proxy-based $\beta_0$ factor. Each $\beta$ model was optimized with respect to deadband type and size before intercomparison.

Concerning deadband form and size, we found an optimum in RMSE for linear deadbands with sizes of 0.5 and $0.9\sigma_w$. These deadband widths make this method approximately equal to the use of a constant $\beta$ factor.

With respect to overall REA performance, we found that the $\beta_w$ and constant $\beta$ from long-term measurements performed more robustly than the proxy-dependent approach $\beta_0$. The latter model still performed well when scalar similarity between the proxy (here sensible heat) and the scalar of interest (here latent heat) show strong statistical correlation, i.e. during periods when the

distribution and temporal behavior of sources and sinks were similar. With respect to sensitivity of $\beta$ to atmospheric stability, we observed that $\beta_0$ slightly increased with increasing stability parameter $z/L$ when no deadband is applied, but this trend vanished with increasing deadband size. $\beta_w$ was independent of $z/L$. To explain these surprising differences, we separated the contribution of w$'$ kurtosis to the flux uncertainty, which can be expressed by the median ratio of the REA flux compared to that from classical eddy covariance $\frac{F_{REA}}{F_{EC}}$. Results showed a strong sensitivity to site conditions: While the kurtosis of w$'$

seems to have no effect on the flux estimate at the grassland site, decreasing trends with increasing kurtosis can be observed





for the loose gravel and forests sites and could explain the variability of $\frac{F_{REA}}{F_{EC}}$ within 10%.

For REA applications without deeper site-specific knowledge of the turbulent transport and degree of scalar similarity, we recommend using either the constant $\beta$ or $\beta_w$ models when REA scalar fluxes are not expected to be limited by the detection limit of the instrument. For conditions close to the instrument detection limit, the $\beta_0$ models using a hyperbolic deadband are the optimum choice.

## 1   Introduction

Trace gases play a significant role in the atmosphere because of their relationship to human-induced climate change, their wide variety of natural and anthropogenic sources, and their impact on human and ecosystem health. Understanding their source and transport behavior is needed to better quantify, predict, and mitigate anthropogenic effects on the environment. The exchange of trace gases between the Earth's surface and the atmosphere is often the result of a combination of several biophysical processes and mechanisms. Observing the net turbulent exchange, i.e. the flux density of such gases can help identifying their sources and sinks, which in turn can help identifying their forcings. Micrometeorological techniques can measure area-integrated fluxes at the ecosystem level and are therefore suitable for computing atmospheric budgets of trace gases and aerosol species.

The most direct method to measure flux density, hereafter referred to as 'flux', between the surface and the atmosphere is the eddy covariance (EC) technique, which requires fast ($\geq 10$ Hz) response sensors to capture all scales of turbulent eddies contributing to the flux. However, such sensors are not available for all trace gases of interest, particularly for reactive species with brief atmospheric lifetimes. In such cases, eddy accumulation (EA) methods, originally proposed by Desjardins (1972, 1977), were developed to provide an alternative means to estimating the net flux. A system of fast switching valves collects air into two separate reservoirs, i.e. one for upward moving eddies (updrafts, $w_+$) and one for downward moving eddies (downdrafts, $w_-$). However, in true eddy accumulation, the number of collected samples must be proportional to the magnitude of the vertical wind speed. For systems with switching valves that are not fast enough to accommodate the shortest time scale of turbulent eddies and/or cannot perform proportional sampling, a relaxation of the original true EA technique is necessary by introducing a proportionality factor. The resulting indirect REA technique, as proposed by Businger and Oncley (1990), thus samples the air with a constant flow rate, dependent on the direction of vertical wind. While the first true EA system is currently under construction (Siebicke, 2016; Siebicke and Emad, 2019), REA approaches are a common and convenient alternative to direct flux measurement of EC and EA when fast-response analyzers for the gas species of interest are unavailable.

For the REA technique, the concentration difference between the two sample reservoirs, $\Delta \overline{c} = (\overline{c_+} - \overline{c_-})$, in which $c_+$ indicates the updrafts and $c_-$ the downdrafts, is linearly related to the vertical net flux $F$ of the species of interest. Due to the sampling relaxation, a linear proportionality factor, usually denoted by the Greek letter $\beta$, is required to compute the flux:

$$F_{REA} = \beta \cdot \Delta \overline{c} \cdot \sigma_w \qquad (1)$$



with $\sigma_w$ being the standard deviation of the vertical wind component $w$. This approach resembles flux-gradient similarity methods evaluated at a single height, where $\beta \cdot \sigma_w$ can be interpreted as an efficiency measure, relating the concentration difference of the scalar of interest to its flux. For practical and scientific reasons, several REA applications exclude samples associated with weak vertical wind speeds that fall into a certain range of values ("band") leading to an unsampled ("dead") region, which effectively acts as a filter (Fig. 1). Deadbands are applied with the intention of (i) increasing the concentration difference between updraft and downdraft reservoirs (Bowling et al., 1998), (ii) to avoid rapid switching between reservoirs due to small

eddies and thus reduce the wear of valves, and (iii) to reduce the random noise in gas concentrations of sampled air, which is mostly due to the small-scale short-lived eddies with a minor flux contribution.

Since the choice of the $\beta$ coefficient and the size and form of the deadband are critical to deriving biophysically meaningful flux measurements from REA, they have received much attention in the literature. Dependency of $\beta$ on the atmospheric stability $z/L$, where L is the Obukhov length (Obukhov, 1946), turbulence and scalar similarity has been discussed, and approaches

including fixed deadbands, constant $\beta$ vs. dynamically adjusting $\beta$ and/or the deadband to atmospheric conditions have been proposed (Businger and Oncley, 1990; Beverland et al., 1996; Katul et al., 1996; Andreas et al., 1998; Milne et al., 1999; Ammann and Meixner, 2002; Fotiadi et al., 2005; Ruppert et al., 2006; Held et al., 2008; Grönholm et al., 2008). The large number of potential combinations for the critical REA parameters and varying site conditions may often seem overwhelming to either the first-time user focusing on investigating the dynamics of a certain trace gas species or even to the advanced user lacking a

detailed understanding of the site-specific turbulent flow conditions. To provide some science-based guidance, our study aims at giving a comprehensive overview covering the most common parameterizations of the $\beta$ factor and the deadband with the goal of providing a practical selection guide for choosing an optimal $\beta$ and deadband model by evaluating them across contrasting ecosystem types. Our choice of contrasting ecosystems is expected to increase the robustness of the findings. We evaluate the $\beta$ models by simulating an idealized REA sampling applied to high time-resolution data of wind components and and

scalar concentrations from field campaigns carried out over contrasting vegetated and non-vegetated surfaces: The McMurdo Dry Valleys of Antarctica, which represent an almost exclusively physically driven ecosystem predominately covered by loose gravel, a biologically active grassland in direct vicinity to agricultural areas in Lindenberg, Germany, and the Waldstein spruce forest site in Germany, where measurements were carried out on a 33-m high tower. 'Idealized' REA sampling here means that any effects of instrument performance are neglected to isolate the flux uncertainty solely related to choosing the critical REA

parameters, i.e. $\beta$ factor, and deadband size and type. We acknowledge that other challenges for measuring trace gas fluxes particularly of reactive components exist, which may substantially add to the uncertainty in REA flux estimates, including low detection limits, high precision demands, and other technical challenges posed by short-lived chemical species. A discussion of these sources of uncertainty are outside the scope of our study. However, even if the latter dominate selecting an optimal $\beta$ model for a specific type of surface can still minimize the overall flux uncertainty.





## 2 Theory of REA and overview of $\beta$ parametrizations

### 2.1 Proxy $\beta_0$ model

The most commonly employed REA variant is based upon scalar-scalar similarity: Observations of a scalar $s$, which is measured with fast-response sensors thus enabling the computation of the direct EC flux $\overline{w's'}$, is used as a proxy for the scalar of interest $c$. The $\beta$ needed for the simulated REA flux of $s$ to equal its measured EC flux is calculated and used for the flux computation of scalar $c$. From this point on, we will refer to $\beta_0$ to represent this proxy approach:

$$\beta_0 = \frac{\overline{w's'}}{\sigma_w \cdot \Delta \overline{s}} \tag{2}$$

where $\Delta \overline{s}$ is the proxy scalar concentration difference between updrafts $(\overline{s(w>0)})$ and downdrafts $(\overline{s(w<0)})$. Often, sonic
temperature $T_S$ is chosen as proxy scalar (e.g. Ren et al., 2011; Osterwalder et al., 2016, 2017) due to its availability and negligible measurement uncertainty. The $\beta_0$ method is based on the strong assumption that the proxy scalar and the scalar of interest behave similarly in their exchange mechanism, which requires the vertical and horizontal distribution of the sinks and sources of both scalars to be identical. A violation of this assumption will inevitably lead to large errors in the REA flux estimate (Katul et al., 1995; Katul and Hsieh, 1999; Ruppert et al., 2006; Riederer et al., 2014). The similarity between the
scalar of interest and the scalar chosen as a proxy can be evaluated by examining the correlation coefficients between the high-resolution time series of two scalars, if available. The scalar-scalar correlation coefficients, $r_{cs}$, as used in other publications (e.g. Gao, 1995; Katul and Hsieh, 1999; Ruppert et al., 2006; Riederer et al., 2014), are defined as follows:

$$r_{cs} = \frac{\overline{c's'}}{\sigma_c \cdot \sigma_s} \tag{3}$$

### 2.2 Vertical wind statistics $\beta_w$ model

An alternative method to determine the $\beta$ factor solely takes into account the standard deviation and difference of the mean vertical wind statistics and was originally derived by Baker et al. (1992). We will refer to this approach as the $\beta_w$ model, in which the proportionality factor is computed using the following relationship:

$$\beta_w = \frac{\sigma_w}{\Delta \overline{w}}. \tag{4}$$

Combining the above equations (1) and (4) yields the expression

$$F_{REA} = \frac{\Delta \overline{c}}{\Delta \overline{w}} \cdot \sigma_w^2 \ = \ m \cdot \sigma_w^2 \tag{5}$$

which does not contain the $\beta$ factor, but instead uses the slope $m$ of the $w'$ vs. $c'$ correlation (Baker, 2000). Compared to the proxy-estimated flux, the scalar flux becomes directly proportional to the vertical wind speed's variance $\sigma_w^2$, and thus to the turbulence statistics. This approach combines elements of the flux-gradient and flux-variance similarity theories. The requirements for this parameterization are (i) a linear relationship between $c'$ and $w'$ through the origin, as well as (ii) the
25 Gaussian distribution of the vertical wind velocity fluctuations. If both are fulfilled, $\beta = 0.63$, however, usually, smaller values





of the $\beta$ parameter are measured (Katul et al., 2018).

The statistical moments of the w′ distribution can be used to investigate deviations from ideally Gaussian. The fourth-order moment of the w′ distribution, i.e. the kurtosis or tailedness, has been explored by Katul et al. (1996), who found an increase of $\beta_w$ with an increasing kurtosis of the w′ distribution. Apart from excursions from an ideal Gaussian w′ distribution, the

c′-w′ correlation also affects $\beta_w$. It was found that large energy-containing eddies (i.e. eddies with large w′) are associated with smaller c′ than predicted by the linear $\Delta\bar{c}$ vs. $\Delta\bar{w}$ fit, resulting in the $\beta_w$ method overestimating the scalar fluxes (Katul et al., 1996; Baker, 2000). Recently, Katul et al. (2018) disentangled effects due to intermittency of the vertical velocity and asymmetry of large coherent structures in w′ during the transport of c′, and were able to explain that $\beta$ is smaller than the theoretical value of 0.63 when taking into account the sweep and ejection phases of coherent structures which are subject to

forcings other than those of the stochastic isotropic homogeneous background turbulence.

### 2.3   Dynamic deadband with constant $\beta_{0,const}$

Grönholm et al. (2008) proposed that a constant value of $\beta$ can be used in REA flux calculation in combination with a dynamic linear deadband scaled by $\sigma_w$. A more detailed description of the deadband application can be found in the subsequent section. The value of $\beta$ is derived by taking the median of $\beta_0$ (of $T_s$ or $CO_2$) over a period of several days:

$$\beta_{0,const} = \hat{\beta}_0, \tag{6}$$

where ˆ represents the median. This method was used e.g. by Osterwalder et al. (2016) to measure mercury fluxes at a peatland site.

### 2.4   Deadband models

Deadbands are widely used in REA applications. The use of a deadband can provide improved resolution in concentration

differences by selectively sampling eddies with a larger contribution to the trace gas exchange. The turbulence characteristics can differ greatly across different ecosystems, therefore, an optimal deadband size must be chosen carefully. In the $\beta_w$ approach, they can limit the impact of weak "distorting" eddies, which contribute little to the flux. Thus, deadbands help improve the linearity of the c′-w′ relationship leading to a well defined $m$ (see Eq. 5).

When applying a linear deadband to w′ (left panel in Fig. 1), no sample is taken if the magnitude of w′ is below a certain

threshold. This threshold can be held constant or adjusted dynamically in time. Dynamical adjustments are often done by scaling with the standard deviation of the vertical wind $a\,\sigma_w$, where $a$ is a constant. This approach offers the advantage of the deadband being proportional to the integral strength of the turbulent diffusive process transporting the trace gas of interest. During field sampling, the size of the deadband is dynamically adjusted by applying a back-looking running window of fixed length to compute $a\sigma_w$. Baker (2000) recommends a linear deadband with a width of $a = 0.9$ to obtain the best estimate of

the slope $m$ in the $\beta_w$ approach.

Hyperbolic deadbands aim to exclude eddies with little flux contribution and maximize the concentration difference between the two sampling reservoirs. The exclusion of up- or downdrafts is in this case not only based on vertical wind velocity, but



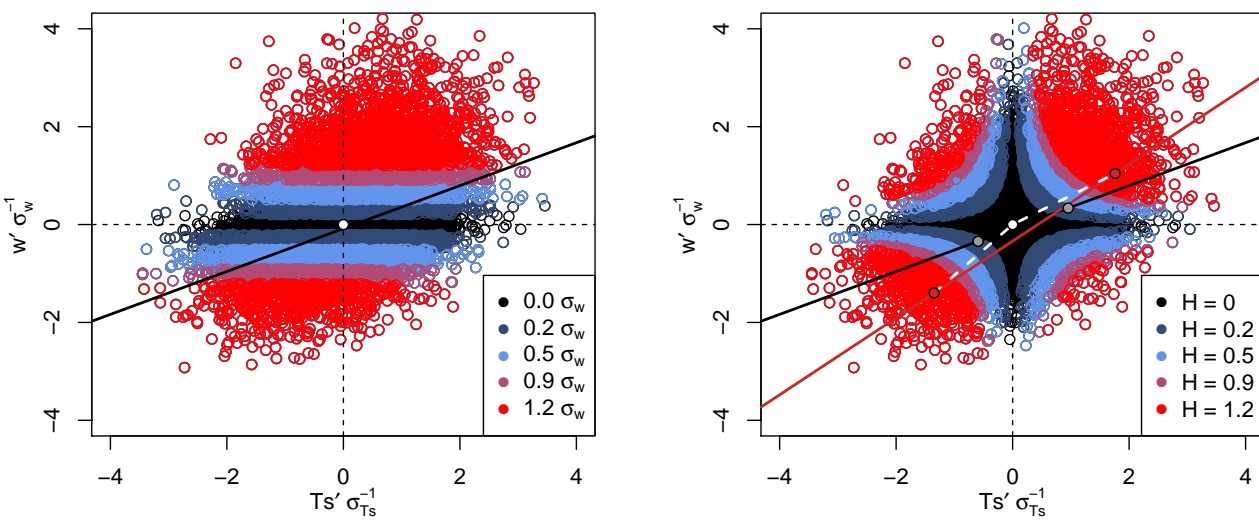

**Figure 1.** Schematic quadrant plots to visualize the application of linear (left) and hyperbolic (right) deadbands. Different colors show which data points are included for different deadband sizes. The white dot marks the origin in both panels. In the right-hand panel, solid grey and red dots mark the mean $w'/\sigma_w$ and mean $T_s'/\sigma_{Ts}$ for up- and downdrafts when no deadband is applied (grey) and when a hyperbolic deadband with $H = 1.2$ is applied (red). The white dashed lines in the right-hand panel connect the red dots with the coordinate system origin. The deviation from $180°$ of the angle spanned between these lines is used as a measure for the asymmetry of the sample distribution.

also on the fluctuations of a proxy scalar. Hyperbolic deadbands are defined by the dimensionless factor $H$, which is defined as (Bowling et al., 1999):

$$H = \frac{\overline{w's'}}{\sigma_w \cdot \sigma_s}.$$
(7)

Plotting a such defined function in velocity-scalar space demonstrates that an area in the shape of two hyperbolas is excluded (right panel in Fig. 1). The REA method using a hyperbolic deadband is often referred to as the HREA technique.

However, the use of large deadbands must be done with caution because they exclude a significant fraction of the data from being sampled, which could lead to increasing the random sampling error caused by a decreasing sampling size, since it is related to $1/\sqrt{n}$. In addition, the time period between opening and closing a sampling valve is reduced for large deadbands, which may introduce additional errors because of the increased difficulty of sampling short-lived events precisely. An estimate for random error can be derived from the asymmetry of the sample distribution. Considering the quadrant plot of the data points sampled in one averaging period, the distribution can be represented by two points: $(\overline{s(w>0)}),(\overline{w(w>0)})$ and $(\overline{s(w<0)})$, $(\overline{w(w<0)})$. Considering the example in Fig. 1, these points are drawn in the right-hand panel for the largest deadband size (red dots) and for the null deadband (black dots). Ideally, these two points fall onto a unique linear relationship intersecting





the coordinate system's origin (white dot). However, the use of large deadbands leads to introducing large asymmetry between up- and downdrafts and deviations from this expectation for the reasons mentioned above. The asymmetry is shown as a white dashed line in the right panel of Fig. 1 containing a bend. This bend, which can be expressed as an angle deviating from $180°$, is used as a measure for the asymmetry of the sample distribution in our study.

### 2.5  Selected models and evaluation metrics

In this study, we compare $\beta$ and deadband approaches used in literature and evaluate their performance for the prediction of
the latent heat flux over different terrestrial surfaces. The following four REA methods have been chosen for the analysis:

–  Model 1: $\beta_0$ using the sensible heat as proxy and a dynamically adjusted linear deadband scaled with $\sigma_w$

–  Model 2: $\beta_0$ using the sensible heat as proxy and a dynamically adjusted hyperbolic deadband scaled with $\sigma_w$

–  Model 3: $\beta_w$ using a dynamically adjusted linear deadband scaled with $\sigma_w$

–  Model 4: $\beta_{0,const}$ (median over the complete field experiments) using sensible heat as a proxy and a dynamically adjusted
linear deadband scaled with $\sigma_w$

For each of the models, five different deadband widths were examined both for linear ($0.2 \cdot \sigma_w$, $0.5 \cdot \sigma_w$, $0.9 \cdot \sigma_w$ and $1.2 \cdot \sigma_w$) and hyperbolic ($H = 0.2$, $H = 0.5$, $H = 0.9$ and $H = 1.2$) deadbands. One simulation was run as a control with a null deadband. To dynamically adjust deadband size, back-looking windows of 60 s and 300 s durations were tested. Comparison of these two window sizes yielded only negligible difference between the computed fluxes for the three datasets, hence we chose to present
results from the 300 s window only.

We proceeded in the following fashion: each of the above models is first optimized with respect to the deadband size. To do so, the accuracy of each $\beta$ model is assessed by comparing the median ratio of the modeled flux, $F_{REA}$, and the corresponding direct EC-measured flux, $F_{EC}$, $\frac{F_{REA}}{F_{EC}}$: If this ratio is greater than 1, then the flux is overestimated by the model, and if it is $< 1$, the flux is underpredicted. In addition, the variability of this ratio is inferred using the root mean square error (RMSE), which
provides a measure of the precision of each model. It is computed using the difference between the modeled REA flux and the measured EC flux:

$$RMSE = \sqrt{\frac{\sum\limits_{i=1}^{n}(F_{REA,i} - F_{EC,i})^2}{n}} \tag{8}$$

The deadband width, which is found to yield the most accurate latent heat flux (taking into account relative error and RMSE), is then further evaluated. Table 2 summarizes the different tested model setups along with the optimal deadband sizes.



**Table 1.** Description of the three data sets used in this study, numbers from quality-screened data, aggregated to 30 min temporal resolution

| surface type | Grassland | Loose gravel | Spruce forest |
|---|---|---|---|
| **site** | Falkenberg site, Lindenberg, Germany | McMurdo Dry Valleys, Antarctica | Waldstein site, ridge in the Fichtelgebirge Mountains in Bavaria, Germany |
| **measurement period** | 2015-09-22 to 2015-10-01 | 2012-12-26 to 2013-01-26 | 2016-06-18 to 2016-07-17 |
| **lat/lon** | $52.17°$N $14.12°$E | $77.57°$S $163.48°$E | $50.13°$N $11.87°$E |
| **elevation above sea level (a.s.l.) [m]** | 73 | 35 | 775 |
| **estimated surface roughness length ($z_0$)[m]** according to Panofsky (1984) | 4.46 | 0.14 | 65.45 |
| **dynamic stability range ($zL^{-1}$)** | -1.95 to 9.38 | -9.61 to -0.01 | -6.74 to 3.96 |
| **horizontal wind speed U [ms$^{-1}$]** | 0.06 to 3.67 | 0.11 to 6.44 | 0.32 to 6.50 |
| **IQR of friction velocity u$_*$ [ms$^{-1}$]** | 0.047 to 0.205 | 0.126 to 0.256 | 0.371 to 0.730 |

**3 Sites and data processing**

**3.1 Site descriptions**

We selected three sites with strongly contrasting vegetation cover and surface roughness, vegetation architecture, and biogeochemical processes governing the vertical exchange of $CO_2$, water vapor and sensible heat to test the different $\beta$ models (Table 1). Using sites with stark differences provides robust recommendations for REA users choosing an optimal $\beta$ for their site.

The grassland data (Thomas et al., 2020) was collected in Falkenberg, Germany at the German Meteorological Service (Meteorological Observatory Lindenberg), (see Table 1 for details). The central part of the field site is a flat meadow of dimensions
5 150 x 250 m covered by short grass (vegetation height < 20 $cm$). This area is surrounded by grassland and agricultural fields in the immediate vicinity, a small village is situated about 600 m to the SE, and a small, but heterogeneous forest area lies to the west and northwest at about 1 to 1.5 km distance. Within the flux footprint of the tower, the main vegetation cover consisted of grassland and recently harvested maize. The soil type distribution in the area around Lindenberg is dominated by sandy soils



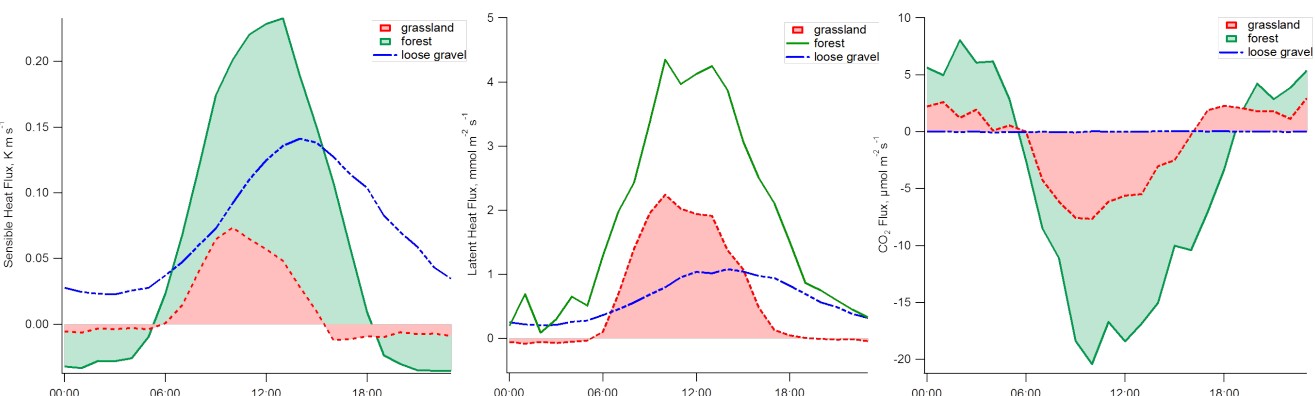

**Figure 2.** Ensemble-averaged diurnal fluxes of sensible heat (left panel), latent heat (center panel) and $CO_2$ (right panel) at each of the three sites. The traces where the sign of the flux changes are filled to the zero line for clarity.

covered by a layer of loam, which is typically found at a depth of between 50 cm and 80 cm. Lindenberg represents moder-

10   ate mid-latitude climate conditions at the transition between marine and continental influences. Monthly mean temperatures (1961-1990) vary between 1.2°C (January) and 17.9°C (July), and the mean annual precipitation sum is 563 mm (Beyrich et al., 2002; Neisser et al., 2002).

In contrast, the McMurdo Dry Valleys (Thomas and Levy, 2020) span 4800 km$^2$ of ice-free land in Antarctica and are covered by rocks and glacial till (Linhardt et al., 2019). The area ranges from sea level to 2000 m in elevation composed of ice-covered lakes, short-lived streams, and rocky ice cemented soils that are surrounded by glaciers. During calm conditions, the region is dominated by a strong near-surface temperature inversion. Strong katabatic winds draining the polar plateau frequently disrupt this inversion. The mean annual temperature in the Dry Valleys ranges between -17 and -20°C. The low precipitation relative

to potential evaporation, low surface albedo, and dry katabatic winds descending from the Polar Plateau result in extremely arid conditions (Clow et al., 1998).

Finally, we use a data set acquired at a spruce forest site in the German Fichtelgebirge (Thomas and Babel, 2020) that spans ca. 1000 km$^2$ of north-eastern Bavaria, Germany. Its summits reach 1053 m a.s.l. (Schneeberg) and 1023 m a.s.l. (Ochsenkopf). The Waldstein hillsides comprise a mountainous ridge reaching up to 877 m a.s.l. (Gerstberger et al., 2004). The measurement

site is located at about 800 m a.s.l.. The prevailing tree species at the Waldstein site is Norway spruce (*Picea abies, L.*) with a mean canopy height of 19 m. The flux measurements were conducted on top of a 31-m high scaffolding tower reaching above the highest tree tops, resulting in a total measurement height of 33 m above ground. Monthly mean temperatures (1961-1990) vary between -4.2°C (January) and 14.1°C (July), and the mean annual precipitation sum is 1156 mm (Foken, 2003).

Comparison of the heat and $CO_2$ fluxes of the three ecosystems highlights the diverse exchange behavior trace gases can

exhibit depending on the environments (Fig. 2). The differences in albedo and length of daylight are reflected in the variation in the sensible heat fluxes. The heat flux at the loose-gravel covered site in the Dry Valleys of Antarctica particularly stands out due to the perpetual sunlight experienced during the campaign period. A diel course is still observed, but is constantly directed





away from the surface indicated by the positive values. The diel patterns seen in both forest and grassland sites are similar, showing positive sensible heat flux during daytime and negative at nighttime. The difference in flux magnitude between forest

and grassland can be attributed to the distinct differences in vegetative canopy properties. The tall dark canopy with low albedo surface is considerably different from the shorter and more reflective grassland canopy. The range in latent heat and $CO_2$ fluxes displays the impact of the vegetation. The loose-gravel site, which is the most extreme site void of vegetation shows a net exchange of $CO_2$ equal or close to zero between the surface and the atmosphere, where both forest and grassland sites display an expected pattern of dominant $CO_2$ uptake during the daytime and respiration during the nighttime. The larger leaf area index in the forest of around 5 $m^2m^{-2}$, compared to 3 $m^2m^{-2}$ for typical grassland areas, like in Lindenberg, causes a greater magnitude of latent heat flux because of the transpiration and concurrent greater exchange of $CO_2$.

## 3.2 Instrumentation and post-field data processing

The turbulence observations consisted of the three-dimensional wind vector and sonic temperature collected by a sonic anemometer (Lindenberg: CSAT3, Campbell Scientific Ltd., Logan, UT, USA; Dry Valleys: 81000 VRE, R.M. Young Company, Traverse City, Michigan, USA; Waldstein forest: USA-1 FHN, Metek, Elmshorn, Germany) and water vapor and carbon dioxide molar densities measured by an open path analyzer (LI-7500, LI-COR Inc., Lincoln, NE, USA) both recorded by a data logger

(CR3000, Campbell Scientific Ltd., Logan, UT, USA). The sampling rate was 20 Hz. Spikes and outliers in raw turbulence time series were discarded according to Vickers and Mahrt (1997) with an initial $5\sigma$ criterion. Resulting gaps in the high-frequency time series were linearly interpolated. Covariances were maximized by shifting the scalar time series relative to that of the vertical velocity by a dynamically determined lag. For the Reynolds decomposition, a perturbation and averaging time scale of 300 s was chosen. Using this shorter than the common 30 min time scale is motivated by the intention to filter out the effects of

longer-lived motions, as described in Vickers et al. (2009). Raw velocities were rotated using the first two steps in the common 3D rotation method ensuring that the mean cross-wind and vertical wind components equal zero. A spectral correction was applied to EC fluxes following Moore (1986) to account for flux losses resulting from the sensor design and data collection. Quality assurance and quality control flags were applied to the computed REA and EC fluxes by testing for stationarity and developed turbulence following Foken et al. (2004). All data with flags $> 1$ were discarded from subsequent analysis. Since

the flags do not capture all unphysical flux estimates, additional hard thresholding was applied. To minimize the substantial random error in turbulent flux estimates over short averaging intervals, six subsequent 300 s intervals were block-averaged to yield one 30 min flux estimate for both the REA and EC method following Vickers et al. (2009).

Since simulating REA sampling requires selecting individual high frequency data from a continuous time series and computing density-corrected scalar higher-order moments, an *ad-hoc* density correction was applied to the water vapor and carbon

dioxide molar densities (Detto and Katul, 2007) prior to flux computations. To this end, molar densities were multiplied by the ratio of the instantaneous to mean density of moist air $\rho_q\overline{\rho_q}^{-1}$. EC fluxes were computed using the common *post-hoc* density correction (Webb et al., 1980). Even though open-path observations in cold environments such as the McMurdo Dry Valleys suffer from sensor heating artifacts not captured by either our *ad-hoc* or the common *post-hoc* WPL correction (Burba et al., 2008), we decided to not apply this additional correction in this study since we are interested in the relative flux error





$(F_{REA} - F_{EC})F_{EC}^{-1}$ only. Instead, we applied a constant offset of 0.35 $\mu$mol m$^{-2}$s$^{-1}$ to the CO$_2$ flux densities to force it through zero for illustrative purposes. This choice has no effect on the study results.

For the REA flux estimation, hyperbolic and linear deadbands of varying sizes were tested. The linear deadband size was scaled by increasing fractions of $\sigma_w$ computed over a back-looking running window of length 300 s (e.g., Ren et al., 2011; Arnts et al., 2013; Movarek et al., 2014). It must be noted that the deadbands are applied only to the $w'$-$c'$ statistics to compute slope $m$ (see Eq. (5)). In contrast, the entire population of vertical velocities observed in an averaging period were used to compute $\sigma_w^2$. Applying the deadbands for computing also the vertical velocity variance leads to significant flux overestimation since $\sigma_w$ increases with increasing deadband size.

This study evaluates estimates of the latent heat flux $\overline{w'q'}$ obtained using different REA techniques. The approaches requiring
a proxy scalar rely on the sensible heat flux $\overline{w'T'}$, which is common choice since it can be measured with a higher precision compared to e.g. CO$_2$ in certain low-flux conditions.

## 4  Results and Discussion

We structured this section to first present scalar correlation coefficients for the different ecosystems in section 4.1. In section 4.2, we describe the choice of an optimal deadband size for each REA model, based on both $\frac{F_{REA}}{F_{EC}}$ and the RMSE. The
optimized REA models are then evaluated in section 4.3 with respect to the effects of the diurnal course and atmospheric stability.

### 4.1  Scalar similarity across land surfaces

Since scalar similarity is an important assumption for the $\beta_0$ models (models 1 and 2) and as evaluation metric for our inter-comparison, we first present the results on its temporal dynamics across land surfaces. To assess whether a scalar can serve
as a viable proxy for the trace gas of interest, the similarity in source and sink strength of two can be represented by their correlation coefficient r$_{x,y}$ (Ruppert et al., 2006). The diurnal courses of the correlation coefficients of $\overline{w'c'}$ and $\overline{w'T'}$, $\overline{w'c'}$ and $\overline{w'q'}$ and $\overline{w'T'}$ and $\overline{w'q'}$, ensemble-averaged over the complete field campaigns, are presented in Fig. 3. Pronounced temporal changes in scalar similarity within the diurnal cycle at the grassland and forest sites are in strong contrast to the constant values observed r$_{x,y}$ in the Dry Valleys (Fig. 3). The patterns can be explained by the influence of radiative forcing, which governs both the physical heat exchanges and biological photosynthesis and evapotranpsiration, highlighting the constant daylight observed during the measurement period in the Dry Valleys. All three correlation coefficients change sign at the grassland site
around 14:00 local time, associated with the expected change in dynamic stability resulting from the change in the direction of the sensible heat flux. A similar diurnal pattern is observed in the forest site, however, the change in sign of the correlation coefficient happens approximately two hours later in the day. In contrast, the correlation between $\overline{w'c'}$ and $\overline{w'q'}$ is positive throughout the nocturnal period in the forest site and negatively correlated in the grassland site, where regular dew formation occurs (which can be also observed in Fig. 2). The scalars tend to be poorly correlated at nighttime compared to daytime as a
result of weak turbulence and associated diminished scalar transport efficiency for both sites.

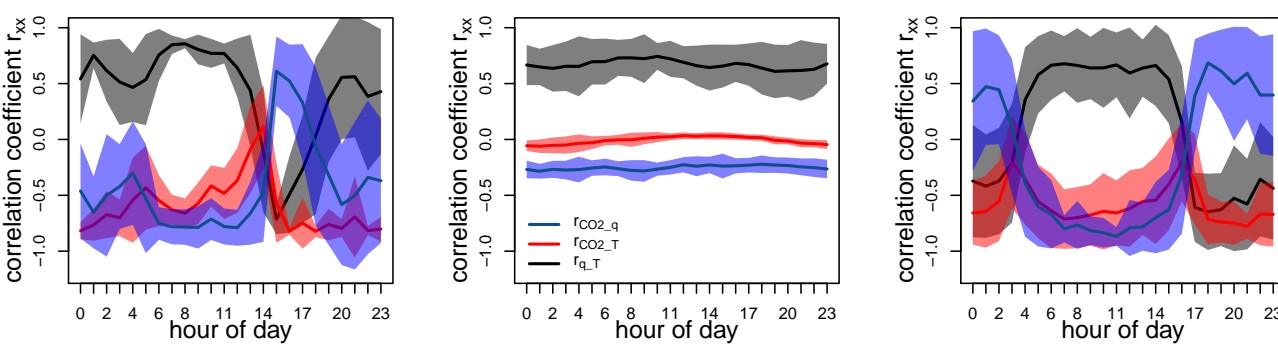

**Figure 3.** Diurnal course of scalar-scalar correlation coefficients $r_{x,y}$ at the meadow site (left panel), the loose gravel-covered Dry Valleys site (center panel) and the Waldstein forest site (right panel). $\pm$ one standard deviation $\sigma$ is drawn as a semi-transparent area around the mean curves of $r_{x,y}$.

## 4.2 Determining the optimal deadband size for each of the $\beta$ methods

As expected, the water vapor concentration difference between up- and downdraft reservoirs $\Delta\overline{q}$ increases with increasing deadband size, for both linear and hyperbolic deadbands, at all three sites. This increase is more pronounced for hyperbolic deadbands compared to the linear deadbands (Fig. 4a and b). The hyperbolic deadbands have the desired effect of maximizing

the concentration difference between the two sampling reservoirs. For $H = 1.2$, almost a factor of 3 increase in water vapor concentration difference between updraft and downdraft sampling reservoirs can be achieved. The HREA technique therefore has the potential to provide concentration differences detectable by instrumentation with high detection limits or when measuring chemical species with very low mixing ratios. However, as mentioned above, large deadbands can introduce a large random error because they exclude a large portion of the sample data points. The decrease in sample size with increasing deadband

size is similar across all three sites (Fig. 4c and d) and should be considered when choosing an optimal deadband. For example, for a hyperbolic deadband with $H = 0.2$, approximately 40% of the sampling period is excluded, which results in an increased asymmetry (Fig. 4d). This effect is more pronounced for the forest and meadow surfaces than for the gravel site, possibly caused by a larger heterogeneity in scalar sink and source distribution.

In the next step, we evaluate each REA model individually and select an optimal deadband size with respect to selected un-

certainty metrics of the $\beta$ model. We chose to include both the precision and the accuracy of the methods by comparing $\frac{F_{REA}}{F_{EC}}$ and RMSE for the simulated deadbands. The ratio $\frac{F_{REA}}{F_{EC}}$ and RMSE obtained for different linear deadband widths using the $\beta_0$ model (model 1) are shown in Fig. 5. Results strongly vary with ecosystem type: While the REA water vapor flux at the Antarctic gravel site is very similar to that obtained from the EC technique, shown by negligible RMSEs, the estimates obtained at the forest and meadow sites have a much larger RSME. This difference can be explained by the differences in the degree

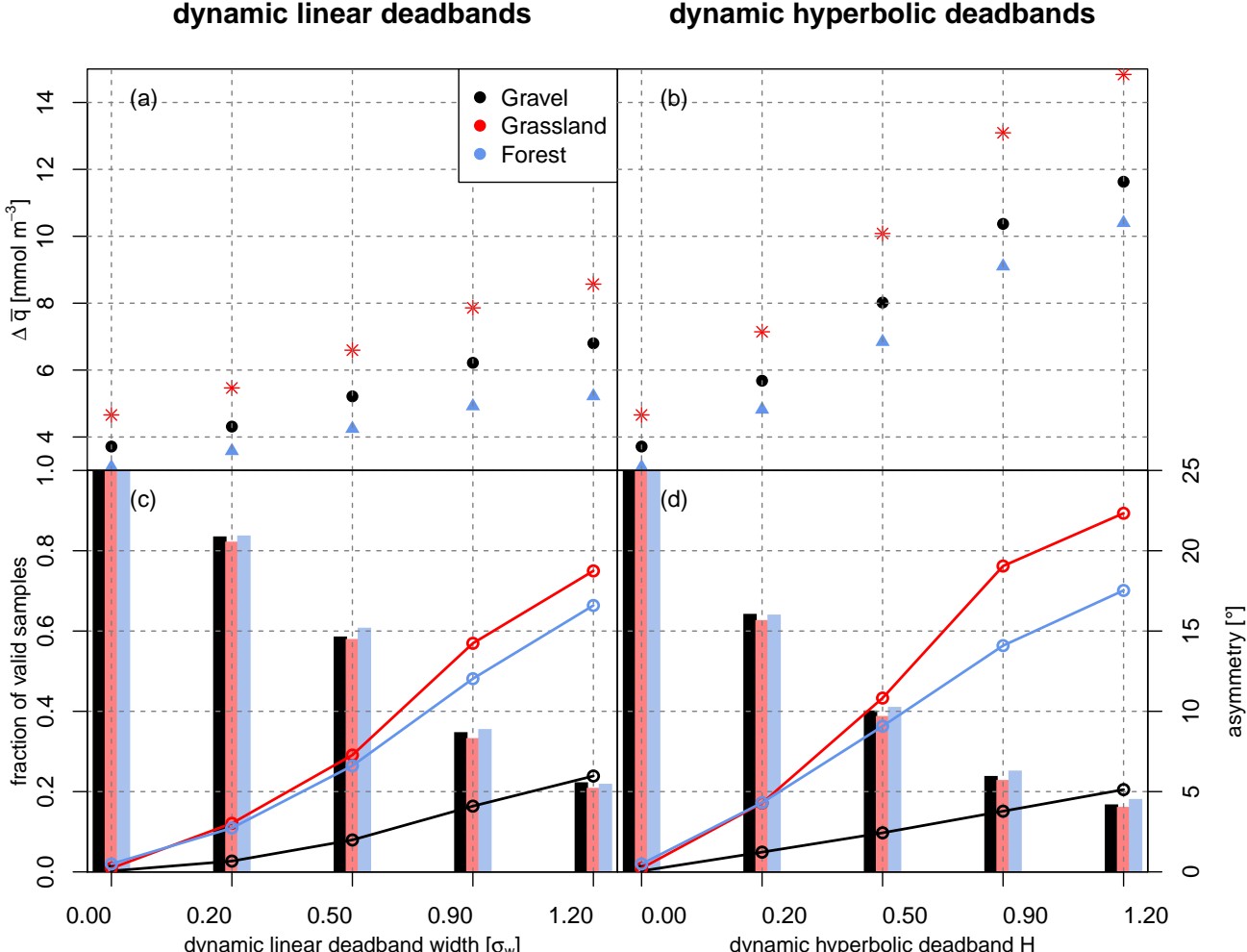

**Figure 4.** Effect of deadband size on the concentration difference and measures of the random error. The top two panels show the concentration difference between up- and downdraft reservoir as a function of deadband size (linear deadbands in left and hyperbolic deadbands in right panel); bottom panels: The bars show the fraction of samples used for flux computation and the overdrawn circles show the asymmetry between up- and downdraft vector in the quadrant plot. The asymmetry is calculated as the absolute deviation in the angle between a straight line and a bent line constructed using the center points of up- and downdrafts and the origin in the quadrant plots, see Fig. 1 and accompanying text for details.





## Dynamic linear deadband with β₀ (model 1)

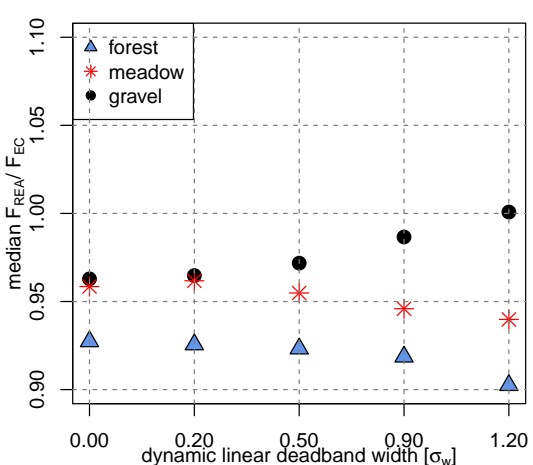 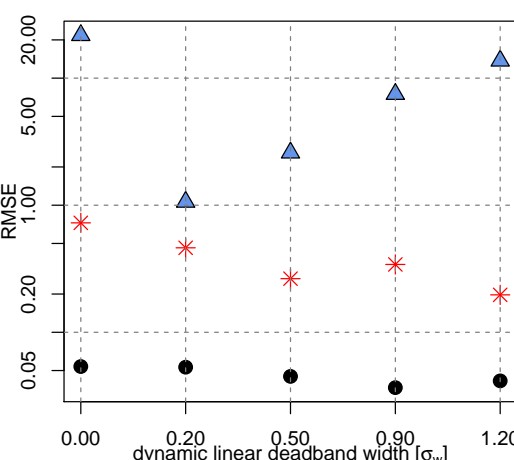

**Figure 5.** Errors as a function of dynamic linear deadband width. The x axis is the scaling factor multiplied with the vertical wind standard deviation $\sigma_w$ to define the deadband threshold. Left panel: Median $F_{REA}/F_{EC}$ (latent heat flux simulated with sensible heat as a proxy) ratio for each of the simulated dynamic deadband widths; right panel: RMSE for each of the simulated dynamic deadband widths

of scalar-scalar similarity between the latent and sensible heat fluxes of the purely physically driven site as opposed to the biologically active sites. The scalar fluxes are modulated by a varying degree of vegetation responses adding to the complexity of the scalar-scalar correlation $r_{q,T}$ and diurnal changes in sign (Fig 3). The use of no deadband (deadband width = 0) leads to an overall small underestimation of the EC fluxes (4 to 8%) across all sites. This underestimation is reduced with the use of a deadband at the gravel site; however, the systematic bias is not resolved by applying a deadband at the other two sites, but on the contrary increases this underestimation. Such a systematic bias could in theory be corrected for in post-processing, but the magnitude of the correction would have to be determined for each site defying our intention of providing general recommendations. This flux bias varies more between sites than with deadband width, therefore, this correction method should only be applied if the user knows the transport characteristics and scalar sink and source distribution well. Based on the flux bias and RMSE, a linear deadband with $0.2\,\sigma_w$ width is chosen as the optimized deadband size for further comparisons with the linear $\beta_0$ approach.

The results for the $\beta_0$ model using a hyperbolic deadband (model 2) are shown in Fig. 6. Both median $F_{REA}/F_{EC}$ and RMSE are of the same order of magnitude compared to the linear deadband approach for $\beta_0$ (Fig. 5, model 1). However, the hyperbolic deadband offers an increase in concentration difference that is considerably larger compared which led to its use in several studies (e.g. Held et al., 2008; Movarek et al., 2014). Interestingly, the observed underestimation of the latent heat flux is lessened for the forest and gravel sites when hyperbolic deadbands are applied, whereas it becomes larger for the meadow site. For the gravel site, the bias even changes sign for large hyperbolic deadbands. The RMSE shows no significant improvement




## Dynamic hyperbolic deadband with $\beta_0$ (model 2)

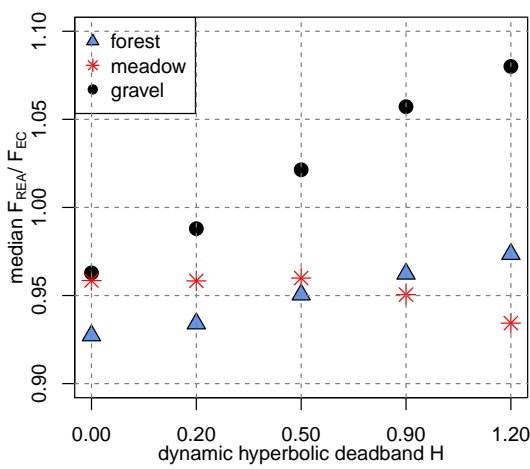
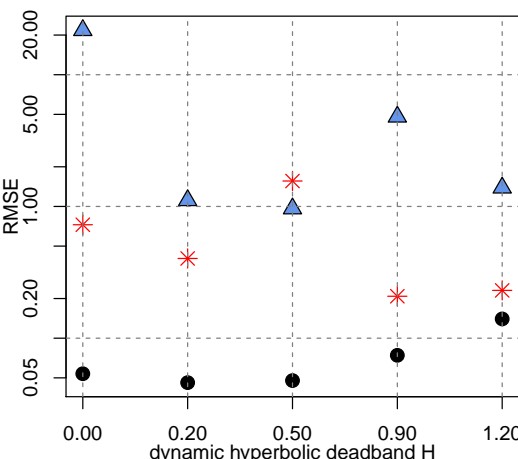

**Figure 6.** Errors as a function of dynamic hyperbolic deadband width. The x axis is the H parameter in Eq. 7, which defines the deadband size. Left panel: Median $F_{REA}/F_{EC}$ (latent heat flux simulated with sensible heat as a proxy) ratio for each of the simulated dynamic deadband widths; right panel: RMSE for each of the simulated dynamic deadband widths

when the small-scale eddies with small flux contributions are excluded irrespective of ecosystem. Based on Fig. 6, a hyperbolic deadband of H = 0.5 is chosen for further analysis, as it offers an increase of $\Delta\overline{q}$ by 100% (Fig. 4).

15     When applying a linear deadband to model 3 using $\beta_w$ derived from the wind statistics alone, a positive flux bias (5-10%) became evident when no deadband is applied (Fig. 7). This observation confirms the findings of Katul et al. (1996): eddies characterized by a large vertical perturbation (w') are known to contain smaller perturbations in sensible heat ($T'_s$) than predicted by the linear fit of $\Delta\overline{w}$ and $\Delta\overline{T_s}$, whose slope is dominated by the many small-scale eddies characterized by a greater $T'_s$. The use of deadbands puts more weight on large eddies, thus deadbands are convenient to improve the estimate of the w' vs. $T'_s$ resulting in a smaller slope $m$ for this model. The choice of deadband size has clear implications in how well the $\beta_w$ model performs. Note the different y-axis scales used in the right-panel graph of Fig. 7 compared to Figs. 5 and 6. The RMSE

5    for this method is roughly two orders of magnitude smaller compared to values observed in the $\beta_0$ models. Overall, the pattern in relative and absolute error is more consistent for the $\beta_w$ model across the three ecosystems compared to the $\beta_0$ models. The optimal deadband width 0.9 $\sigma_w$, which was proposed by Baker (2000), provides low systematic bias, high precision, and the minimum in RMSE for all sites in Fig. 7. However, the use of this deadband size excludes more than 60% of the available data (see Fig. 4), so we chose a linear deadband width of 0.5 $\sigma_w$ instead. This choice yields a similarly high accuracy and precision

10    and therefore was our optimal choice for model comparisons.

    The performance of the constant $\beta_{0,const}$ (model 4) is illustrated in Figure 8 , in which the $F_{REA}/F_{EC}$ and RMSE was calculated using a constant $\beta$ and dynamic linear deadbands of different sizes. Its RMSE is similar to that of the $\beta_w$ (model





## Dynamic linear deadband with $\beta_w$ (model 3)

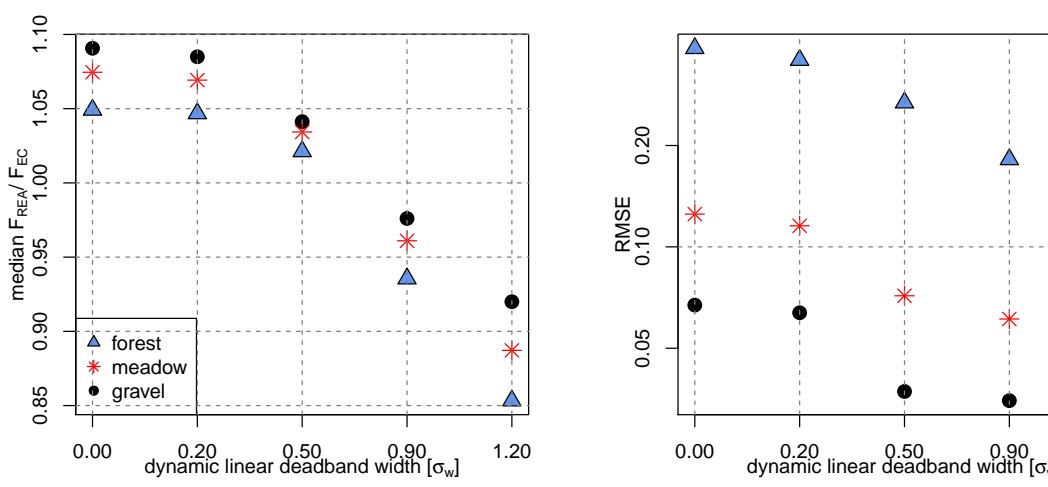

**Figure 7.** Errors as a function of dynamic linear deadband width. The x axis is the scaling factor multiplied with the vertical wind standard deviation $\sigma_w$ to define the deadband threshold. Left panel: Median $F_{REA}/F_{EC}$ ratio (latent heat flux simulated using the REA approach described in Baker (2000)) for each of the simulated dynamic deadband widths; right panel: RMSE for each of the simulated dynamic deadband widths

3) (Fig. 7) with both in the order of magnitude and behavior with increasing deadband widths for all ecosystems. Following Grönholm et al. (2008), we chose a deadband size of 0.5 $\sigma_w$ for further comparison.

### 4.3 Evaluation of optimized $\beta$ models

After choosing an optimal deadband size for each REA model, we now proceed to analyzing the effects of the diurnal light variability and atmospheric stability on flux estimates.

#### 4.3.1 Effect of the diurnal course

5 Data were binned according to the hour of day and the RMSE was computed for each hour. Each panel in Figure 9 shows the result for the different $\beta$ models at the optimal deadband size. All four REA models successfully capture the flux at the loose gravel site, however, discrepancies between $F_{REA}$ and $F_{EC}$ become obvious for the meadow and forest sites. Figure 9a and b show the RMSE of the two $\beta_0$ models using linear (model 1) and hyperbolic (model 2) deadbands. The error for these two methods is significantly larger compared to the REA methods which are plotted in the bottom two panels: The constant

10 $\beta_{0,const}$ (model 4) and the $\beta_w$ method (model 3), both utilizing a dynamic linear deadband, feature a negligible RMSE for the gravel and the meadow sites, and a small RMSE below unity for the forest site. The $\beta_0$ models have a distinct peak in





### Dynamic linear deadband with constant $\beta_0$ (model 4)

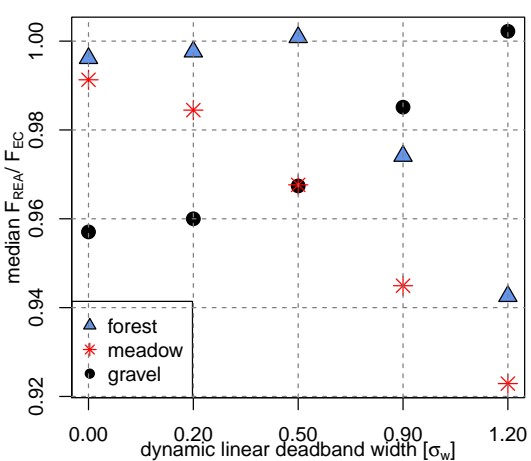
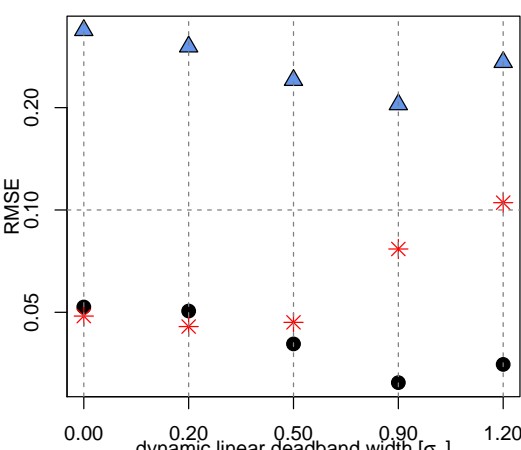

**Figure 8.** Median $F_{REA}/F_{EC}$ ratio as a function of dynamic linear deadband width. The x axis is the scaling factor multiplied with the vertical wind standard deviation $\sigma_w$ to define the deadband threshold. Left panel: Median $F_{REA}/F_{EC}$ ratio (latent heat flux simulated using constant $\beta$ and dynamic linear vertical wind deadband) for each of the simulated dynamic deadband widths; right panel: RMSE for each of the simulated dynamic deadband widths

RMSE at the meadow site around 14:00 local time, which coincides with low scalar-scalar correlation of water vapor and heat (Fig. 3). At the forest site, the uncertainty in $F_{REA}$ is large throughout the diurnal course for both $\beta_0$ models due to the large variability in $r_{q,T}$. During times of strong variability, the difference $F_{REA}$ - $F_{EC}$ can be on the same order of magnitude as the absolute evapotranspiration. This occasional poor performance of the $\beta_0$ model does not change significantly across the range of tested linear and hyperbolic deadbands. Filtering out small-scale eddies therefore does not improve flux estimates. However, hyperbolic deadbands still increase the concentration difference $\Delta \bar{c}$ if the detection limit is of concern. Applying the REA proxy model for observing the diurnal variation of the exchange of a trace gas must be done carefully when choosing a proxy

5  that exhibits a pronounced diurnal cycle. The key assumption in this approach is that the proxy and trace gas of interest have similar temporal or spatial dynamics, which introduces large uncertainties if the temporal dynamics of the scalar of interest remain unknown or are not known a priori.

### 4.3.2 Effect of atmospheric stability

10  The observed relationship between the $\beta_0$ model bias and changes in scalar-scalar similarity suggests a dependence from shortwave radiative forcing leading to changes in atmospheric dynamic stability. A dependence of the $\beta_0$ factor on atmospheric



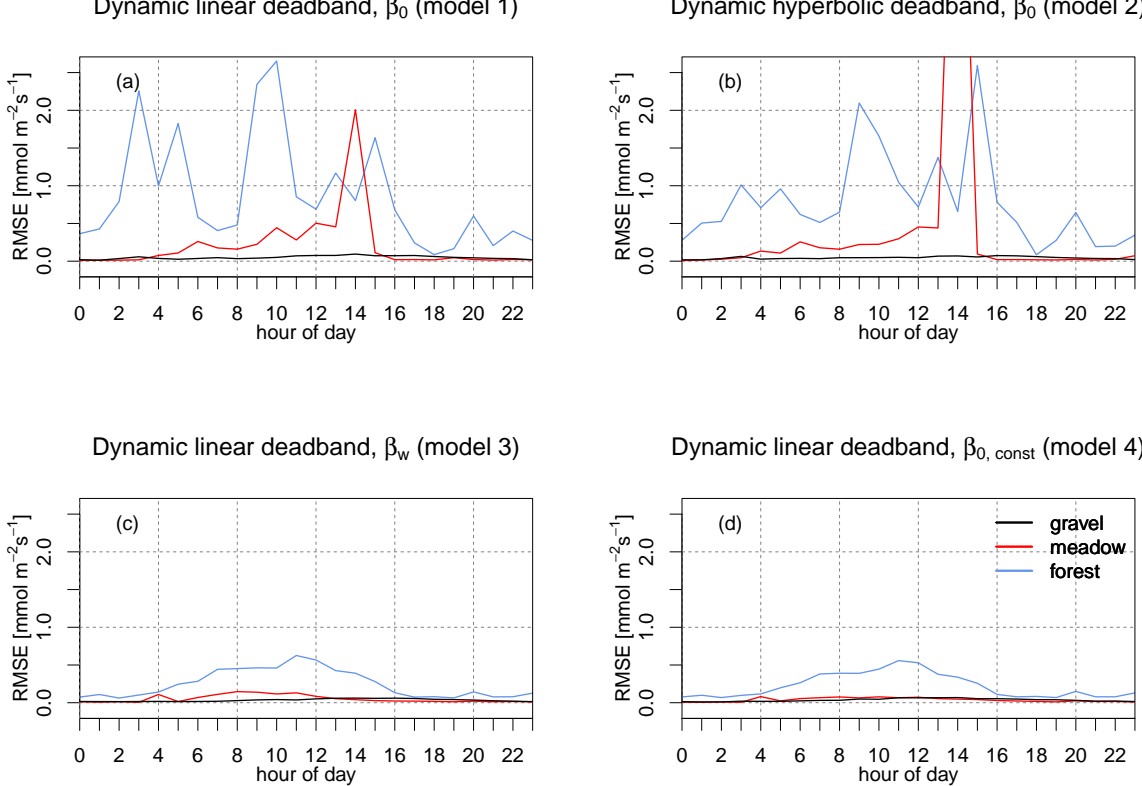

**Figure 9.** Flux RMSE as a function of the hour of day (local time) for each of the optimized $\beta$ models. (a) shows the RMSE of the proxy-based model using a dynamically adjusted linear deadband (model 1), scaled with $0.2\sigma_w$. (b) shows the proxy-based model using a dynamically adjusted hyperbolic deadband (model 2) with $H = 0.5$. In this panel, there is one extreme value with RMSE > 3, which is not plotted. (c) and (d) show the RMSE of the $\beta_w$ REA model (model 3), and the constant $\beta_{0,const}$ approach (model 4), respectively, both with a dynamically adjusted linear deadband of width $0.5\sigma_w$.

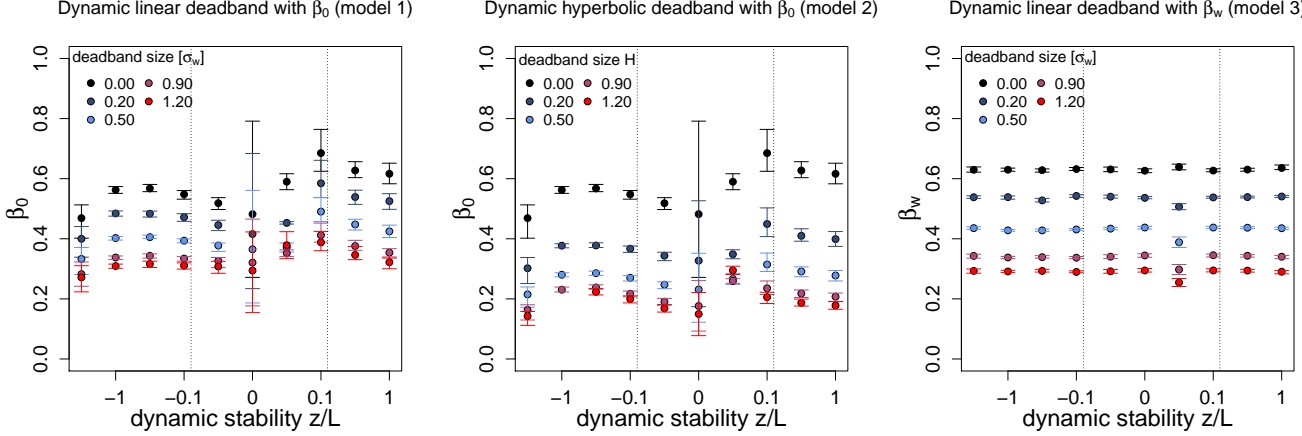

**Figure 10.** Dependence of the $\beta_0$ and $\beta_w$ factors on the atmospheric stability z/L. Data were binned in logarithmic evenly spaced stability classes. The markers are drawn at the median $\beta$ of each bin, the arrows mark the inter-quartile range (IQR). The vertical dashed lines mark the boundaries of the neutral stability class, i.e. for the markers drawn at x=0. All other stability bins are spaced logarithmically in smaller intervals.

stability has been shown in previous studies. Here, we extend this analysis to the effects of deadband type and size in addition to the $\beta_w$ method. For comparison reasons, we evaluate the dependence of the $\beta$ on dynamic stability (z/L) using bins identical to those in Ammann and Meixner (2002) (their Fig. 3), who first documented a relationship between $\beta_0$ and atmospheric stability. Figure 10 shows the models for time-varying $\beta$ binned into logarithmically spaced classes of dynamic stability. The neutral stability class (markers drawn at z/L = 0) spans a larger stability range indicated by the two dashed vertical lines in each panel. For large deadbands, $\beta$ decreases as the scalar concentration difference between up- and downdraft reservoir increases. For the two proxy models (left and center panel in Fig. 10), $\beta_0$ follows the relationship found by Ammann and Meixner (2002) of a

5 constant $\beta_0$ for unstable conditions, and an increase from neutral and stable conditions of $z/L \geq 0.06$. However, this increase is associated with large statistical uncertainty. We therefore recommend exercising caution when using stability-dependent parameterizations of $\beta_0$ for neutral and stable conditions. Ammann and Meixner (2002) analyzed data without a deadband (indentical to our deadband size $\sigma_w = 0$ and $H = 0$ in the left and center panels of Fig. 10). A similar trend is observed for the models with linear and hyperbolic deadbands. Variability of $\beta_0$ generally decreases with increasing deadband width. The

10 $\beta_w$ model (right panel in Fig. 10) shows significant differences compared to the proxy approaches: $\beta_w$ does not change with dynamic stability and is less variable compared to that of $\beta_0$, as seen in the narrower spread of the arrows defining the inter-quartile range. This finding explains why the results in Figures 5, 8 and 9 for the $\beta_w$ and constant $\beta$ applying a dynamic linear deadband are so strikingly similar: $\beta_w$ for the selected optimal deadband width of $0.5\sigma_w$ shows little variability, which makes this approach similar to applying a constant $\beta$ factor.

As pointed out in previous REA studies, $\beta_w$ scales with the fourth central statistical moment of the vertical velocity perturba-





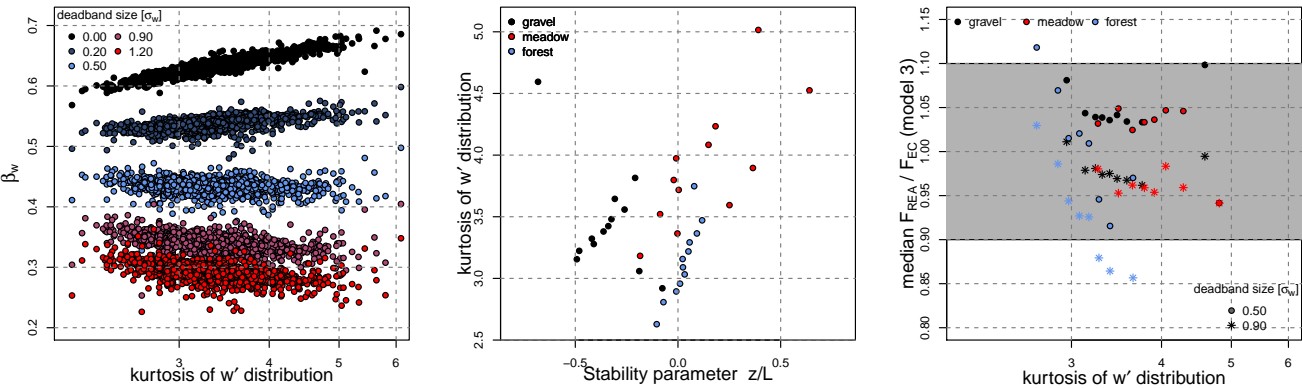

**Figure 11.** Left panel: $\beta_w$ as a function of w′ kurtosis for different deadband widths (not binned). Center panel: the stability parameter z/L as a function of the w′ kurtosis. Data were binned by z/L prior to plotting and only bin medians are displayed. Right panel: Median $F_{REA}$ by $F_{EC}$ as a function of w′ kurtosis for the optimal deadband widths, $0.9\,\sigma_w$ and $0.5\,\sigma_w$, which were determined by Baker (2000) and in this study. Data were grouped into kurtosis bins.

tions' distribution by altering the w′ vs. c′ relationship. We therefore investigated the impact of the w′ kurtosis on the $\beta_w$ factor for different linear deadband sizes. Katul et al. (2018) found that two different factors, which both depend on $z/L$, contribute to $\beta_w$ and whose impacts can cancel out if their magnitudes are similar. The first effect, leading to an decrease of $\beta_w$ with increasing $z/L$, depends on the excess kurtosis, or flatness factor of the w′ distribution. The second effect, resulting in an increase of $\beta_w$ with increasing $z/L$, is a result of the transport efficiency $e_T$ (Wyngaard and Moeng, 1992), as well as source strength and asymmetry in the w′ distribution. The superimposition of these two processes could be an explanation why there is no clear dependence of $\beta_w$ of dynamic stability visible in Figure 10. The relationship between the w′ distribution's kurtosis

and the $\beta_w$ factor is illustrated in Figure 11: consistent with Katul et al. (1996, 2018) the $\beta_w$ factor without deadband increases as a function of w′ kurtosis (left panel). The plot collapses data from all three ecosystems onto a single linear relationship. This finding suggests that the turbulence statistics are ubiquitous despite the significant differences in climate and surface characteristics across the three ecosystems. The increasing linear trend becomes less pronounced when deadbands are applied. For the deadband size of $0.5\,\sigma_w$, no trend was found.

Kurtosis is in turn related to dynamic stability, as can be seen in the center panel in Fig. 11: Changes in turbulence statistics with changing diabatic conditions lead to non-Gaussian distribution of w′. As a result, the kurtosis of the w′ distribution becomes different from 3, which is the prediction for a Gaussian distribution. The right panel of Fig. 11 displays the median $\frac{F_{REA}}{F_{EC}}$ as a function of w′ kurtosis. Only the model results for REA applying a linear deadband with widths of 0.5 and $0.9\,\sigma_w$ are displayed for improved visibility. While no clear trend was observed at the grassland site, and only a slightly negative trend was observed at the gravel site, we found a strong decrease of the median $\frac{F_{REA}}{F_{EC}}$ as a function of w′ kurtosis for the forest.

However, as is indicated by the shaded area in the rightmost panel of Fig. 11, most points lie within the boundaries of $\pm\,10\%$.





Only the bins with the highest and lowest kurtosis classes at the forest site are outside of this range. These error bounds are of the magnitude as the error assumed in EC applications. We suspect that the large excursions from Gaussian statistics for the forest site are caused by coherent structures forcing cross-canopy vertical exchange, which are a dominant flow mode in the forest flows documented for this site (Thomas and Foken, 2007a, b).

## 5  Conclusions and practical recommendations

In this study, we compared the performance of four different models to compute the scalar latent heat flux from conditional sampling simulations. The tested REA models included the following methods: Two proxy-based approaches using (i) linear and (ii) hyperbolic deadbands; (iii) a parameterization of the $\beta$ factor first introduced by Baker et al. (1992) and (iv) an approach using a constant $\beta$ factor described in Grönholm et al. (2008). Table 2 summarizes the REA models investigated along with the main results of this study. The proxy-based ($\beta_0$) REA models performed well during conditions when the proxy scalar (here sensible heat) and scalar of interest (here latent heat) were strongly correlated, i.e. during periods when sources and sinks were similarly spatially distributed and temporally synchronized. Median ratios $\frac{F_{REA}}{F_{EC}}$ over the campaign length were close to unity, indicating a generally high accuracy of the methods. The diurnal course of the flux bias showed large deviations from the EC flux, particularly during transitions when the direction of the flux changed. This increased variability vanishes when REA fluxes are ensemble-averaged over several diurnal cycles. However, users are strongly cautioned when using the $\beta_0$ proxy models. The diurnal dynamics of the proxy scalar and trace gas of interest is of central importance. This is also true for scalars subject to both biological and physical forcings driven by time- and space-variant source-sink distributions. Choosing the optimal proxy scalar is critical for the method's success. Hyperbolic deadbands, which also require the use of a proxy scalar, are well suited to maximize the concentration difference between up- and downdraft reservoirs more effectively than linear deadbands. The effects of linear and hyperbolic deadbands on the flux estimates were strongly site-dependent for the proxy-based approaches. For the $\beta_w$ and constant $\beta_{0,const}$ models, an optimum size was found for 0.5 and $0.9\sigma_w$. In general, these two models performed more robustly than the proxy-dependent approaches. The RMSE of these methods, which are utilizing dynamically adjusted $\sigma_w$-dependent linear deadbands, was orders of magnitude lower.

The dependence on atmospheric stability conditions was evaluated for each method. No universal behaviour of any stability-dependent ($z/L$) $\beta$ model for either site was observed. We therefore cannot recommend its use.

*Data availability.*  data sets are soon available at Zenodo: Thomas and Levy (2020), Thomas et al. (2020), Thomas and Babel (2020)

*Author contributions.*  CKT led the field experiments and performed the REA flux computations. TV and AH performed the data analysis and visualization. TV, AH and CKT wrote the manuscript.





**Table 2.** Summary of the REA models compared in this study, along with main findings

| model # | model 1 | model 2 | model 3 | model 4 |
|---|---|---|---|---|
| summary description | $\beta_0$ + linear deadband | $\beta_0$ + hyperbolic deadband | $\beta_w$ + linear deadband | constant $\beta$ + linear deadband |
| optimal deadband size | $0.2 \cdot \sigma_w$ | $H = 0.5$ | $0.5 \cdot \sigma_w$ | $0.5 \cdot \sigma_w$ |
| site-specific effects | magnitude of underestimation differs between sites | positive or negative bias differs between sites | no strong site-dependent implications found; intermittent turbulence could have a weak effect on the accuracy of the method | no strong site effects |
| accuracy | within 8% | within 5% | within 5% | within 5% |
| precision | RMSE $\approx$ $1mmol\ m^{-2}s^{-1}$ for biologically active sites | RMSE $\approx$ $1mmol\ m^{-2}s^{-1}$ for biologically active sites | RMSE below $0.3mmol\ m^{-2}s^{-1}$ | RMSE below $0.3mmol\ m^{-2}s^{-1}$ |
| diel dependency | strong correlation with $r_{xx}$ | strong correlation with $r_{xx}$ | none | none |
| remarks | | recommended when detection limit is an issue | robust method | robust method |

*Competing interests.* The authors declare that no competing interests are present.

*Acknowledgements.* We would like to thank the German Meteorological Service (DWD) for granting access to their Falkenberg site at the Lindenberg observatory. We further express our gratitude to Wolfgang Babel and Johannes Olesch for their assistance in collecting the observations at the Lindenberg and Waldstein sites in Germany, and Joseph Levy for the opportunity of and assistance in collecting observation in the Dry Valleys of Antarctica. CKT acknowledges funding from the National Science Foundation Career Award in Physical and Dynamical Meteorology, award AGS 0955444. TV would like to thank the R project (R Core Team, 2015), especially the developers of the *chron* (James and Hornik, 2017), *xts* (Ryan and Ulrich, 2014) and *classInt* (Bivand, 2015) packages for providing free data analysis and visualization tools. We also acknowledge the valuable feedback of two anonymous reviewers on a previous manuscript draft.



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
