# Peer review of "Choosing an Optimal $\beta$ Factor for Relaxed Eddy Accumulation Applications Across Vegetated and non-Vegetated Surfaces"

_Biogeosciences, 2020_

## Referee Comment (RC1) · Anonymous Referee #1 · 24 Jan 2021

The authors present an evaluation of four different REA $\beta$-factor estimation approaches. They evaluate the four approaches for the $H_2O$ flux at three different sites with very different vegetation cover (forest, meadow, gravel). The comparison of the different approaches and the different sites are the main advantage of the study. Specifically, the performance of the $\beta_w$ approach is included (which is rarely reported in the literature) and for the proxy $\beta$ approach, the use of an overall constant $\beta$ value is compared to a half-hourly adjusted value. A main result of the study is that the use of a constant $\beta$ value per site (or individual $\beta_w$ values, which show a quasi-constant behavior) is superior to the use of half-hourly determined proxy $\beta$ values.

However, the evaluation of REA approaches is less comprehensive than declared in the objectives. Only one scalar ($H_2O$ / latent heat) is used for the approach validation, and only one scalar (T) is used as proxy. Moreover, the manuscript suffers from a number of additional shortcomings that need some substantial improvements before publication. They are listed in the following comments.

*Important*: The line numbering of the manuscript is erroneous (non-sequential) on most pages, which made the review somewhat cumbersome. I use the true text line numbers in the following comments (not the ones indicated in the manuscript).

MAJOR COMMENTS

1) Only the performance of the REA approaches for the $H_2O$ flux is tested in the present study. This is done after an initial deadband optimization (using the reference EC dataset) for the same test scalar. This leads to a certain lack of independence in the method validation. Although the $CO_2$ flux and its correlation with the other scalar fluxes is introduced in Sections 3 and 4.1, the REA evaluations for the CO2 flux are unfortunately not presented. Alternatively $CO_2$ could have served as second proxy scalar option beside the temperature T (at least for some sites) as indicated in Section 2.3.
The authors should more prominently (in abstract and objectives) declare that they are evaluating the REA approaches only for $H_2O$ fluxes. In addition they need to discuss better, whether and why they assume that the results also apply to other scalars, despite a sometimes low scalar correlation as exhibited in Fig. 3.

2) I find it a bit misleading to use the index "0" for the $\beta$ factor of the proxy scalar approach. Obviously (see scalar correlation analysis) it matters, which scalar is used as proxy. Therefore, it would be more informative and more consistent to use the scalar specific index "T" or "wT" for the proxy scalar approaches here.

3) The presentation of the $\beta_w$ approach in Section 2.2 is a bit confusing in my view. It is not clear what the use of Eq. 5 is for a REA application. The factor "m" is a purely theoretical quantity that has no use for practical REA applications. Therefore the practical $\beta_w$ approach evaluated in the present study should be clearly separated from theoretical considerations.
In addition, the alternating use of "$\beta$" and "$\beta_w$" in this section is confusing. E.g. it is argued (P5, L5) that "the c'-w' correlation also affects $\beta_w$". But this is contradicting the definition of $\beta_w$ (Eq. 4) purely depending on the w-distribution.

4) How can it be that the zero deadband calculations result in RMSE of about 20 mmol $m^{-2}$ $s^{-1}$ for the forest site in Figs. 5 and 6, when the fluxes themselves are only between 0 and 4 mmol $m^{-2}$ $s^{-1}$ (Fig. 2) and the flux ratios in the left panels are close to 1? This seems very unplausible and needs a detailed explanation.

5) The resulting $\beta_{0,const}$ values and the average $\beta_w$ values for the three different sites should be listed in a Table, so that other researchers can compare them to their own results.

6) For Figure 10 and 11 it is not indicated, which data are displayed. Are these all (valid) data for all three sites or only data from one site? This needs to be clearly stated in the Figure caption.

7) I have some problems when comparing the $\beta_w$ results displayed in Fig. 10 (right panel) and Fig. 11. The zero deadband results in Fig. 11 show a considerable variation with the kurtosis and that the kurtosis systematically depends on stability. In contrast the $\beta_w$ results in Fig. 10 show practically no variation, neither with stability nor within the bins.

MINOR COMMENTS

P1, L15-16: It is not clear, which $\beta$ approach this sentence is related to.

P2, L2-5: Both sentences are formulated in a misleading way.
First sentence: the detection limit of the instrument does not limit the REA fluxes directly but the quality/uncertainty of the REA fluxes. Change e.g. to "...when the uncertainty of the REA flux quantification is not limited by ...". Second sentence: change to: "For REA sampling differences close to the instruments detection limit ..."

P6, Fig. 1: Please check if the position of the grey points in the right panel is correct. According to Eq. 4 and a $\beta_w$ value of about 0.6, the normalized vertical distance of the two grey points (=$\Delta w / \sigma_w$) should be about 1.6, but in the figure this distance is much less than 1. Maybe the x- and y-axis need to be exchanged...?

P8, Table 1: The units for the roughness length are probably [cm], not [m]. Only indicate two significant digits in the roughness length values, because their accuracy is not so high.
Please also include the average canopy heights and the EC measurement heights in the table (better than scattered in the text). This would be advantageous for the reader.

P9, L17: The formulation "...resulting in a total measurement height ..." is not logical (what results in what?). Please rephrase.

P9, last line: Correct to: "A diel course is still observed, but the flux is constantly directed ..."

P10, L18: What do you mean with "perturbation time scale"

P10, L25: Explain the "additional hard thresholding".

P11, Line 1-2: I do not understand what "force it through zero" means here.

P11, Line 5: I do not understand why the slope m had to be computed in the present study. It is not necessary for the $\beta_w$ calculation according to Eq. 4. Moreover, the w'-c' statistics are not available in a real REA application (see also comment 3 above).

P20 Fig. 11 middle and right panel: The symbol colors hardly distinguishable. Removing the black frame of the symbols may be helpful.

P21, L7-10 ("The tested REA models .... along with the main results of this study.") This part should be omitted from the Conclusions because it is pure repetition.

Figures 5-8: Indicate the units of the RMSE in the right panels.

Table 2: In the second lowest row, "$r_{xx}$" presumably should be replaced by "$r_{xy}$"

---

## Referee Comment (RC2) · Anonymous Referee #2 · 22 Feb 2021

This paper presents a contribution to the evaluation of the $\beta$ parameter required for the relaxed eddy-accumulation (REA) technique. This technique is used to measure land-atmosphere exchange of scalars for which analyzers fast enough to implement the eddy-correlation technique are not available. The study is based on fast observations of temperature, vertical wind and humidity, on three contrasting ecosystems during a few weeks. The authors have simulated a relaxed eddy-accumulator on the recorded time series, and have compared the resulting moisture flux estimated for different $\beta$ models with the eddy-correlation value, the latter being considered as the true value of the flux.

[Figure]

General comments:

1. I am not convinced that BG is a suitable journal for such a study. The paper is technical, and does not offer any process analysis. In my opinion, AMT would be more appropriate. But I leave to the Editor(s) the settlement of this question.

2. There is an abundant literature on REA, $\beta$ determination and sensitivity to various parameters. By the way the authors mention numerous previous studies in their paper. However, they do not clearly indicate what is really innovative in their study, what is a progress with respect to previous estimations/models, etc.. For example, the detection limit and sensitivity of the analyzers is often an obstacle for trace species flux estimates. The authors indicate in their abstract that "For conditions close to the instrument detection limit, the $\beta 0$ models using a hyperbolic deadband are the optimum choice.", but this statement is not really supported by a study in which time series would have been degraded to simulate a less performing analyzer.

3. The authors evaluate several models for the parameter $\beta$. It is sometimes difficult while reading the paper to clearly understand to what model it is referred to. For example, it is written in the abstract "We tested a total of three different REA models for the $\beta$ factor...", whereas in the text 4 models are analyzed. In section 2.5, when the 4 models are presented, the corresponding relevant equations should be recalled. Furthermore, since they are numbered (#1... #4), the reference to the corresponding number should be systematically given both in the text and the figures.

4. The paper is confusing in several parts regarding the use of density vs. mixing ratio to express concentration. This is an important question, since we know from 40 years that density fluctuations have a considerable impact on flux estimates. This question is as crucial for REA as for eddy-correlation fluxes. See also my specific comments relative to this question below.

5. The authors present $CO_2$ fluxes in their set of observations, but they do not use them to evaluate the $\beta$ models. Only water vapour fluxes are analyzed. Why?

Specific comments and drafting matter:

1. P. 1, line 22: "To explain these surprising differences,...". To what differences is it referred to?

2. P. 2, lines 15-16: EA or REA techniques are NOT adapted for highly-reactive species, because concentrations might evolve under chemical reactions occurring during the accumulation period of time. For such species, disjunct eddy covariance technique can offer an interesting alternative.

3. P. 2, line 26: The term "concentration" is ambiguous. It must be clearly indicated whether it means "density" (expressed in e.g. kg/m3, or mol/m3) or "mixing ratio" (expressed in e.g. kg/kg or mol/mol). Note that the mixing ratio value is conserved when temperature, pressure or density vary, which is not the case for density. When using an REA system, the mixing ratios have to be measured in the reservoirs at the end of the accumulation period because densities can have been modified under the variation of temperature and pressure conditions in the reservoirs. Similarly, if c is expressed as a mixing ratio, the correct form for equation (1) should involve the mean air density (see e.g. Bowling et al. 1999).

4. P. 4, equation (4): $\Delta w$ is not defined.

5. Section 2.4: A discussion about the symmetrical vs. asymmetrical deadbands is missing.

6. P. 5, line 19: "applying a linear deadband to w'": please explain what is meant by "linear".

7. P. 5, line 22: "the deadband being proportional to the integral strength of the turbulent diffusive process". This is unclear. What is the "integral strength"?

8. P. 5, lines 26-27: "Hyperbolic deadbands aim to exclude eddies with little flux contribution and maximize the concentration difference between the two sampling reservoirs.". This is not specific to hyperbolic deadbands since the same can be said regarding constant deadbands.

9. Figure 1: I suggest to plot the symbols with a single colour, and add lines of different colours to indicate the thresholds, rather than plotting couloured symbols. As it stands the Figure is ambiguous: when we look at, for example, the light blue 0.5 $\sigma$w threshold, all the red and brown dots should also be included in this class.

10. Figure 1: What represent the black lines?

11. Equation (1): Defined in that way, H is simply the correlation coefficient between w and s. The correct expression is to remove the overbar in (1) and mention that only the samples for which abs(H) is higher than a given threshold are retained.

12. Equation (1) and throughout the text: "c" is used for an unspecified variable, but later on it represents the CO2 concentration. This is confusing.

13. Page 6, line 5: what is "n"?

14. Page 7, last line of 2.4: I am not convinced that the angle reflects the asymmetry of the sample distribution. A highly-skewed sample distribution would be represented by dots at a really different distance from the (0,0) coordinate origin, but these dots could be aligned with the origin.

15. Table 1: I suppose there is a mistake in the unit of roughness length.

16. Fig. 2: Replace "sensible heat flux" with "kinematic heat flux" and "latent heat flux" with "moisture flux"; or convert units into Wm-2.

17. Page 9: "During calm conditions, the region is dominated by a strong near-surface temperature inversion.". This is surprising, because Fig. 2 shows a positive (upward) heat flux, which is in conflict with a near-surface temperature inversion.

18. Page 9: "Strong katabatic winds draining the polar plateau frequently disrupt this inversion.". This is not consistent with the wind values given in Table 1.

19. Page 10, line 13: Please explain what is a "dynamically determined lag.".

20. Page 10, line 20: Please explain what is meant by "additional hard thresholding was applied.".

21. Page 10, lines 25-27 (also in line with comment#4 above): "To this end, molar densities were multiplied by the ratio of the instantaneous to mean density of moist air q <q>−1. EC fluxes were computed using the common post-hoc density correction (Webb et al., 1980).". This is unclear. Please check carefully. When one computes eddy-correlation fluxes with the scalar fluctuation expressed in mixing ratio unit (or any other proportional unit), the so-called WPL correction is not to be done. What I understand here is that the authors start with a mixing ratio, convert it to a quantity proportional to a density, and eventually apply the WPL correction (one step back, one step forward...).

22. Figure 4: "valid samples" is not the most appropriate terminology. Rather "selected samples", or something equivalent...

23. Figure 5: Why a logarithmic scale on the right panel ? RMSE unit is missing.

24. Figure 10: It is unclear how the stability bins are defined. It seems that the neutral class is very large, encompassing stable and unstable conditions until abs(z/L)∼0.1. There is therefore an evidence of overlapping between the classes.

25. Figure 10, caption: "bars" instead of "arrows".

26. Figure 11: It is not clear which $\beta$ model is represented here. Explain in the caption what is the grey zone.

27. Page 20: "increasing z/L" is ambiguous since z/L could be either positive or negative.

28. Page 2; line 9: replace "sensible heat" and "latent heat" with "temperature" and "water vapour", respectively.

29. Page 21: "methods. The diurnal course of the flux bias showed large deviations from the EC flux, particularly during transitions when the direction of the flux changed". This is unclear, please rephrase.

30. Conclusion: A common name should be used for each model between the text and in Table 2.

31. Section "Conclusions and practical recommendations": The last sentence of the abstract contains a recommendation which is not present here.

---

## Author Comment (AC1) · 28 Apr 2021

**Reply to comments of Reviewer # 1:**

We want to thank the anonymous referee for carefully reading our manuscript and for their helpful comments. We have organized the reviewer comments in a manner such that RxCn represents the nth comment of referee x, and RxSn the nth specific comment by reviewer x. We hope this will provide a clear basis for discussion during the further reviewing process. We are addressing the raised comments in a point-by-point way below:

The authors present an evaluation of four different REA β-factor estimation approaches. They evaluate the four approaches for the $H_2O$ flux at three different sites with very different vegetation cover (forest, meadow, gravel). The comparison of the different approaches and the different sites are the main advantage of the study. Specifically, the performance of the $\beta_w$ approach is included (which is rarely reported in the literature) and for the proxy β approach, the use of an overall constant β value is compared to a half-hourly adjusted value. A main result of the study is that the use of a constant β value per site (or individual $\beta_w$ values, which show a quasi-constant behavior) is superior to the use of half-hourly determined proxy β values. However, the evaluation of REA approaches is less comprehensive than declared in the objectives. Only one scalar ($H_2O$ / latent heat) is used for the approach validation, and only one scalar (T) is used as proxy. Moreover, the manuscript suffers from a number of additional shortcomings that need some substantial improvements before publication. They are listed in the following comments. Important: The line numbering of the manuscript is erroneous (non-sequential) on most pages, which made the review somewhat cumbersome. I use the true text line numbers in the following comments (not the ones indicated in the manuscript)

We are sorry for the additional work generated by erroneous line numbers. Thank you for making the extra effort.

**R1C1)** Only the performance of the REA approaches for the $H_2O$ flux is tested in the present study. This is done after an initial deadband optimization (using the reference EC dataset) for the same test scalar. This leads to a certain lack of independence in the method validation. Although the $CO_2$ flux and its correlation with the other scalar fluxes is introduced in Sections 3 and 4.1, the REA evaluations for the $CO_2$ flux are unfortunately not presented. Alternatively $CO_2$ could have served as second proxy scalar option beside the temperature T (at least for some sites) as indicated in Section 2.3.

The authors should more prominently (in abstract and objectives) declare that they are evaluating the REA approaches only for $H_2O$ fluxes. In addition they need to discuss better, whether and why they assume that the results also apply to other scalars, despite a sometimes low scalar correlation as exhibited in Fig. 3.

The reason why only the results for the $H_2O$ flux are presented was to limit the analysis to a reasonable scope. Additionally, we decided to not present the $CO_2$ flux results because, for the gravel site (Antarctica), there is basically no measurable $CO_2$ flux due to lack of biological activity, which makes the interpretation difficult. However, we agree that, for method validation, considering another flux than the one for which the deadband size was optimized is required. Following the referee's suggestion, we propose adding an appendix (Appendix A), in which we present the hourly binned RMSE evaluation, which was done for $H_2O$ in Fig. 9, but for the $CO_2$ flux. Alternatively, the below figure and interpretation could be included and discussed in the main manuscript. We would like to leave this decision to the editor. Regarding the second part of the comment, we state that the changes will be reflected in abstract and introduction.

[Figure]

Fig. 11: Same as Fig. 9 but for the $CO_2$ flux. The gravel site results (solid black lines) should be regarded with caution as the magnitude of the $CO_2$ flux at this site is close to zero (compare to Fig. 2).

Interpretation: *The same findings that were drawn from the $H_2O$ flux analysis are also apparent in the above figure: Both proxy approaches (panels (a) and (b) ) result in higher values of the RMSE than the $\beta_w$ (panel (c) ) and the constant $\beta$ (panel (d) ) methods. The RMSE for both proxy approaches at the meadow site peaks during 13-14 UTC, the time when scalar-scalar correlation of sensible heat and $CO_2$ is lowest. At the forest site, the RMSE for the $\beta_T$ approaches is highest when the magnitude of the $CO_2$ is largest. The RMSE for the gravel site is included in this figure even though the magnitude of the $CO_2$ flux is close to 0 throughout the daily course and thus no conclusions should be drawn from its RMSE.*

**R1C2)** I find it a bit misleading to use the index "0" for the $\beta$ factor of the proxy scalar approach. Obviously (see scalar correlation analysis) it matters, which scalar is used as proxy. Therefore, it would be more informative and more consistent to use the scalar specific index "T" or "wT" for the proxy scalar approaches here.

We take this comment into account and agree with the reviewer using scalar specific indices are more clear. The adjustments have been made in the figures and the manuscript.

**R1C3)** The presentation of the $\beta_w$ approach in Section 2.2 is a bit confusing in my view. It is not clear what the use of Eq. 5 is for a REA application. The factor "m" is a purely theoretical quantity that has no use for practical REA applications. Therefore the practical $\beta_w$ approach evaluated in the present study should be clearly separated from theoretical considerations. In addition, the alternating use of " $\beta$" and " $\beta_w$ " in this section is confusing. E.g. it is argued (P5, L5) that "the c'-w' correlation also affects $\beta_w$ ". But this is contradicting the definition of $\beta_w$ (Eq. 4) purely depending on the w-distribution.

We strive for our study to be helpful and understandable to users. Therefore, we thank the reviewer for pointing us to these inconsistencies in Section 2.2.

However, regarding the first part of the comment, we would like to argue that one needs to do both: A theoretical introduction, and an evaluation of the practical method. The latter cannot be done without the first, because the practical method needs to rest on a firm theoretical foundation. Our aim is to provide a brief yet comprehensive derivation, starting from theoretical considerations, in our Methods section.

We propose to rephrase the first part of Section 2.2 as follows:

*"An alternative REA method was originally derived by Baker et al. (1992), and Baker (2000) provided a comprehensive derivation. It primarily rests upon the standard deviation of the vertical wind $\sigma_w$, and assumes velocity-scalar correlation. In brief, the flux is defined as:*

$$\overline{w'c'} = m \cdot \sigma_w^2$$

*where m is the regression-estimated slope of the w' vs. c' correlation. m can be approximated, using conditional sampling techniques, as:*

$$m = \frac{\Delta \overline{c}}{\Delta \overline{w}}$$

*This makes:*

$$F_{REA} = \frac{\Delta \overline{c}}{\Delta \overline{w}} \cdot \sigma_w^2$$

*and as a result, a $\beta_w$ factor can be derived as follows:*

$$\beta_w = \frac{\sigma_w}{\Delta \overline{w}}.$$

*The scalar flux becomes directly proportional to the vertical wind speed's variance $\sigma_w^2$, and thus to the turbulence statistics. This approach combines elements of the flux-gradient and flux-variance similarity theories.*

*The requirements for this parameterization are (i) a linear relationship between c' and w' through the origin, as well as (ii) the Gaussian distribution of the vertical wind velocity fluctuations. If both are fulfilled, $\beta_w = 0.63$, however, usually, smaller values of the $\beta_w$ parameter are measured (Katul et al., 2018). "*

**R1C4)** How can it be that the zero deadband calculations result in RMSE of about 20 mmol m-2 s-1 for the forest site in Figs. 5 and 6, when the fluxes themselves are only between 0 and 4 mmol m-2 s-1 (Fig. 2) and the flux ratios in the left panels are close to 1? This seems very unplausible and needs a detailed explanation.

Thanks for spotting this. The large RMSE compared to the median $F_{REA}/F_{EC}$ ratio close to 1 was actually due to one single outlier. We decided to take the physical plausibility thresholds, which were applied to the EC fluxes, and also apply them to all simulated REA fluxes. This removes the outlier in question, and reduces the RMSE values for the forest site in Figs 5 and 6. However, the thresholding does not alter any of the other presented results significantly. The main finding presented in this section, i.e. that the proxy-based approaches result in a larger error compared to the $\beta_w$ and $\beta_{T,const}$ approaches, remains still valid.

We propose to include the following explanation in Section 3.2, stating that the physical plausibility thresholds were applied to the simulated REA fluxes as well:

*"In the final step, the same thresholds for physical plausibility which were applied to the computed EC fluxes were also used to remove unplausible REA flux estimates from the data sets. These thresholds were chosen individually for each scalar and each data set due to the wide range of meteorological and biochemical conditions covered in this study."*

Updated Figs. 5 and 6:

**Dynamic linear deadband with $\beta_T$ (model 1)**

[Figure]

[Figure]

*Fig. 5: Errors as a function of dynamic linear deadband width. The x axis is the scaling factor multiplied with the vertical wind standard deviation $\sigma_w$ to define the deadband threshold. Left panel: Median $F_{REA}/F_{EC}$ (latent heat flux simulated with sensible heat as a proxy) ratio for each of the simulated dynamic deadband widths; right panel: RMSE for each of the simulated dynamic deadband widths*

**Dynamic hyperbolic deadband with $\beta_T$ (model 2)**

[Figure]

[Figure]

*Fig. 6: Errors as a function of dynamic hyperbolic deadband size. The x axis is the H parameter in Eq. 10, which defines the deadband size. Left panel: Median $F_{REA}/F_{EC}$ (latent heat flux simulated with sensible heat as a proxy) ratio for each of the simulated dynamic deadband sizes; right panel: RMSE for each of the simulated dynamic deadband sizes*

**R1C5)** The resulting $\beta_{0,const}$ values and the average $\beta_w$ values for the three different sites should be listed in a Table, so that other researchers can compare them to their own results.

We thank the reviewer for this good suggestion. As recommended, we are adding a table, in which the β values found for each of the sites and methods are listed for the respective optimum deadband sizes:

**Table 2.** median $\beta$ parameters for the chosen optimum deadband sizes for each of the four models and each of the three sites

| site | meadow | gravel | forest |
|---|---|---|---|
| $\beta_w$, **linear deadband width** $\sigma_w = 0.5$ | 0.43 | 0.43 | 0.44 |
| $\beta_T$, **linear deadband width** $\sigma_w = 0.2$ | 0.46 | 0.47 | 0.51 |
| $\beta_T$, **hyperbolic deadband width** $H = 0.5$ | 0.25 | 0.26 | 0.27 |
| $\beta_{T,const}$, **linear deadband width** $\sigma_w = 0.5$ | 0.38 | 0.39 | 0.42 |

We propose to also add the following sentence to the end of Section 4.2:

*„Table 2 summarizes the chosen optimum deadband widths for each of the four methods and gives the medians of the respective β parameters for each of the three sites. "*

**R1C6)** For Figure 10 and 11 it is not indicated, which data are displayed. Are these all (valid) data for all three sites or only data from one site? This needs to be clearly stated in the Figure caption.

Thanks for bringing up this issue. In Figs 10 and 11, all valid data from all three sites are combined. The observations from all three ecosystems fall along the same lines, which suggests that e.g. the findings of $\beta_w$ vs. kurtosis as a function of deadband size presented in the left panel of Figure 11 are ubiquitous.

For clarification, we are adding the following sentence to the caption of Fig. 10:

*"This figure combines valid data points from all three sites."*

and we are adding

*"Valid data points from all three sites are combined in this panel."*,

to the caption of Fig. 11.

**R1C7)** I have some problems when comparing the $\beta_w$ results displayed in Fig. 10 (right panel) and Fig. 11. The zero deadband results in Fig. 11 show a considerable variation with the kurtosis and that the kurtosis systematically depends on stability. In contrast the $\beta_w$ results in Fig. 10 show practically no variation, neither with stability nor within the bins.

We agree, at first sight these observations appear to contradict each other. However, when carefully evaluating these findings, it is an effect of the binning. Fig. 11 center panel only displays the bin medians, while Fig.11 left panel shows the unbinned 30-min data. Comparing the ranges of the w-kurtosis in these panels, one learns that the range between 3 and 4 (center) is much smaller compared to 2 and 5 (left). Within the approximate bounds of 3 and 4 (where most of the data are for all three sites), $\beta_w$ for zero deadband also has a much smaller systematic variability. In combination with Fig. 10 right panel it means is that the bin median value of the w-kurtosis artificially exaggerates the stability dependence, since the within-bin variability is very large, leading to its effect disappearing in the effective $\beta_w$ (Fig. 10, right) and $F_{REA}/F_{EC}$ (Fig. 11, right) findings.

We have added arrows for the IQR of kurtosis and z/L to the center panel of Fig. 11 to make this point more clear, and updated the figure description:

[Figure]

*Fig. 11: Left panel: βw as a function of w' kurtosis for different deadband widths (not binned). Valid data points from all three sites are combined in this panel. Center panel: the stability parameter z/L as a function of the w' kurtosis. Data were binned into eight kurtosis bins with equivalent number of data points. Only bin medians are displayed, arrows mark the IQR. Right panel: Median $F_{REA}/F_{EC}$ as a function of w' kurtosis for the optimal deadband widths, $0.9\,\sigma_w$ and $0.5\,\sigma_w$, which were determined by Baker (2000) and in this study. Data were grouped into the same kurtosis bins as in the center panel.*

We propose to add the above explanation to the text and restructure the combined discussion of Figs. 10 and 11 as follows to make it more logical:

[revised manuscript text omitted]

We want to keep the rather theoretical discussion of w-kurtosis to provide an observation for a potential explanation as to why beta values vary. Again, we believe it is important to understand the theoretical foundation of approaches, irrespective of their impact on the practically applied method.

---

## Author Comment (AC2) · 29 Apr 2021

**Reply to comments of Reviewer # 2:**

We want to thank the reviewer for their careful reading and helpful comments, which clearly helped improve our manuscript. We have organized the reviewer comments in a manner such that RxCn represents the nth comment of referee x, and RxSn the nth specific comment by reviewer x. We hope this will provide a clear basis for discussion during the further reviewing process. We are addressing the raised comments in a point-by-point way below:

This paper presents a contribution to the evaluation of the β parameter required for the relaxed eddy-accumulation (REA) technique. This technique is used to measure land-atmosphere exchange of scalars for which analyzers fast enough to implement the eddy-correlation technique are not available. The study is based on fast observations of temperature, vertical wind and humidity, on three contrasting ecosystems during a few weeks. The authors have simulated a relaxed eddy-accumulator on the recorded time series, and have compared the resulting moisture flux estimated for different β models with the eddy-correlation value, the latter being considered as the true value of the flux.

**R2C1)** I am not convinced that BG is a suitable journal for such a study. The paper is technical, and does not offer any process analysis. In my opinion, AMT would be more appropriate. But I leave to the Editor(s) the settlement of this question.

We appreciate this comment. Our submission to BG was motivated by the journal's focus on the **interaction** between biological, chemical, and physical processes, which is basically the main concern of the flux measurement community. We were hoping to find a good platform to reach the audience interested the most in our results by publishing in BG. We leave the decision to the editor.

**R2C2)** There is an abundant literature on REA, β determination and sensitivity to various parameters. By the way the authors mention numerous previous studies in their paper. However, they do not clearly indicate what is really innovative in their study, what is a progress with respect to previous estimations/ models, etc.. For example, the detection limit and sensitivity of the analyzers is often an obstacle for trace species flux estimates.

The authors indicate in their abstract that "For conditions close to the instrument detection limit, the $\beta_0$ models using a hyperbolic deadband are the optimum choice.", but this statement is not really supported by a study in which time series would have been degraded to simulate a less performing analyzer.

To date, the existing papers investigating the REA method either focus on only one or a smaller selection of β approaches, and/or a limited selection of sites (mostly only one site). These limitations make transferring those results to an arbitrary (new) site difficult, which aggravates the choices a REA user needs to make. To our best knowledge, no study has compared across these different methods. Hence, we included all β approaches across a very broad range of contrasting sites, this is the main innovation and contribution. We are proposing to add the following sentence to the Abstract to make this point clearer:

*"To our best knowledge, this is the first study inter-comparing these different approaches over a range of different sites."*

Furthermore, at least in the atmospheric chemistry community, the $\beta_w$ method is not well known and our contribution demonstrates that it is capable of yielding results as good as (or actually even superior to) the better-known proxy approaches. Another interesting outcome of this study is that the use of the constant $\beta_T$ factor performed better than $\beta_T$ factors which are adjusted for each sampling period.

We are rephrasing part of the Abstract as follows:

*"With respect to overall REA performance, we found that the $\beta_w$ and constant $\beta_{T, const}$ performed more robustly than the proxy-dependent approaches."*

Regarding the second part of the comment, we cannot possibly simulate the variety of analyzers all subject to different detection limits in our analysis. The latter a potential user is most familiar with, but she/he may require guidance on the $\beta$ model. The ultimate choice which uncertainty ($\beta$ approach, or analyzer detection limit) weights more heavily, is up to the user.

**R2C3)** The authors evaluate several models for the parameter $\beta$. It is sometimes difficult while reading the paper to clearly understand to what model it is referred to. For example, it is written in the abstract "We tested a total of three different REA models for the $\beta$ factor...", whereas in the text 4 models are analyzed. In section 2.5, when the 4 models are presented, the corresponding relevant equations should be recalled. Furthermore, since they are numbered (#1... #4), the reference to the corresponding number should be systematically given both in the text and the figures.

The authors agree that the wording in the abstract is misleading. We rephrased the abstract text as follows to make it more consistent:

*"We tested a total of four different REA models for the $\beta$ factor: The first two methods, referred to as model 1 and model 2, derive $\beta_T$ based on a proxy for which high-frequency observations are available (sensible heat $T_s$). In the first case, a linear deadband is applied, while in the second case, we are using a hyperbolic deadband. The third method, model 3, employs the approach first published by Baker et al. (1992), which computes $\beta_w$ solely based upon the vertical wind statistics. The fourth method, model 4, uses a constant $\beta_{T, const}$ derived from long-term averaging of the proxy-based $\beta_T$ factor. Each $\beta$ model was optimized with respect to deadband type and size before intercomparison."*

Furthermore, we want to thank the reviewer for the idea to recall the relevant equations in section 2.5, which will certainly help the comprehensibility of our methods description. We are adding them where we list the different model setups:

"

- *Model 1: $\beta_T$ (Eq. 2) using the sensible heat as proxy and dynamically adjusted linear deadband scaled with $\sigma_w$ (Eq. 9)*

- *Model 2: $\beta_T$ (Eq. 2) using the sensible heat as proxy and dynamically adjusted hyperbolic deadband scaled with $\sigma_w$ (Eq. 10)*

- *Model 3: $\beta_w$ (Eq. 7) using a dynamically adjusted linear deadband scaled with $\sigma_w$ (Eq. 9)*

- *Model 4: $\beta_{T, const}$ (Eq. 8; median over the complete field experiments) using the sensible heat as proxy and dynamically adjusted linear deadband scaled with $\sigma_w$ (Eq. 9)"*

**R2C4)** The paper is confusing in several parts regarding the use of density vs. mixing ratio to express concentration. This is an important question, since we know from 40 years that density fluctuations have a considerable impact on flux estimates. This question is as crucial for REA as for eddy-correlation fluxes. See also my specific comments relative to this question below.

We want to thank the reviewer for raising this important point. We are addressing this issue in more detail below (see our answer to comment **R2S3**).

**R2C5)** The authors present $CO_2$ fluxes in their set of observations, but they do not use them to evaluate the β models. Only water vapour fluxes are analyzed. Why?

This issue was raised by the anonymous referee #1 as well, and was addressed in the response to **R1C1**. In short, $CO_2$ was excluded to keep the scope small, and because there is basically no detectable $CO_2$ flux at the Antarctic site. We are presenting a short evaluation of simulated $CO_2$ fluxes in the response to **R1C1** and propose to add it as an appendix, or include it into the Results section.

Here is the response to **R1C1**:

The reason why only the results for the $H_2O$ flux are presented was to limit the analysis to a reasonable scope. Additionally, we decided to not present the $CO_2$ flux results because, for the gravel site (Antarctica), there is basically no measurable $CO_2$ flux due to lack of biological activity, which makes the interpretation difficult. However, we agree that, for method validation, considering another flux than the one for which the deadband size was optimized is required. Following the referee's suggestion, we propose adding an appendix (Appendix A), in which we present the hourly binned RMSE evaluation, which was done for $H_2O$ in Fig. 9, but for the $CO_2$ flux. Alternatively, the below figure and interpretation could be included and discussed in the main manuscript. We would like to leave this decision to the editor. Regarding the second part of the comment, we state that the changes will be reflected in abstract and introduction.

[Figure]

*Fig. S1: Same as Fig. 9 but for the $CO_2$ flux. The gravel site results (solid black lines) should be regarded with caution as the magnitude of the $CO_2$ flux at this site is close to zero (compare to Fig. 2).*

Interpretation: *The same findings that were concluded from the $H_2O$ flux analysis are also apparent in the above figure: Both proxy approaches (panels (a) and (b) ) result in higher values of the RMSE than the $\beta_w$ (panel (c) ) and the constant $\beta$ (panel (d) ) methods. The RMSE for both proxy approaches at the meadow site peaks during 13-14 UTC, the time when scalar-scalar correlation of sensible heat and $CO_2$ is lowest. At the forest site, the RMSE for the $\beta_T$ approaches is highest when the magnitude of the $CO_2$ is largest. The RMSE for the gravel site is included in this figure even though the magnitude of the $CO_2$ flux is close to 0 throughout the daily course and thus no conclusions should be drawn from its RMSE.*

**Specific comments and drafting matter:**

**R2S1)** P. 1, line 22: "To explain these surprising differences,...". To what differences is it referred to?

Thanks for spotting this, we understand that our text reads a little incoherently here. We refer to the difference between the $\beta_w$ and $\beta_T$ approaches with respect to their dependence on atmospheric stability. We propose to rephrase this part of the abstract as follows:

*"To explain why the $\beta_w$ method seems to be insensitive towards changes in atmospheric stability… "*

**R2S2)** P. 2, lines 15-16: EA or REA techniques are NOT adapted for highly-reactive species, because concentrations might evolve under chemical reactions occurring during the accumulation period of time. For such species, disjunct eddy covariance technique can offer an interesting alternative.

We agree, however, DEC only offers an advantage for highly reactive species if the residence time in the system is small. While we think that adding a DEC simulation would be outside the scope of this paper, we agree that this method should be acknowledged in the introduction in order to give a more comprehensive overview. We are rephrasing the relevant part of the introduction as follows:

*"However, such sensors are not available for all trace gases of interest, particularly for reactive species with brief atmospheric lifetimes. In these cases, Disjunct Eddy Covariance (DEC, Rinne and Ammann, 2012), i.e. non-continuous sub-sampling of the concentration and wind data series, offers one alternative to overcome this problem. Eddy Accumulation (EA) methods provide another solution for estimating the net flux of chemically more stable atmospheric species existing at very low concentrations. This technique was originally proposed by Desjardins (1972, 1977): In EA, a system of fast switching valves collects air into two separate reservoirs..."*

**R2S3)** P. 2, line 26: The term "concentration" is ambiguous. It must be clearly indicated whether it means "density" (expressed in e.g. $kg/m^3$, or $mol/m^3$) or "mixing ratio" (expressed in e.g. kg/kg or mol/mol). Note that the mixing ratio value is conserved when temperature, pressure or density vary, which is not the case for density. When using an REA system, the mixing ratios have to be measured in the reservoirs at the end of the accumulation period because densities can have been modified under the variation of temperature and pressure conditions in the reservoirs. Similarly, if c is expressed as a mixing ratio, the correct form for equation (1) should involve the mean air density (see e.g. Bowling et al. 1999).

We agree, the entire volume expansion/ contraction argument leads to a significant correction term incorporated into the WPL correction (see Detto, M., & Katul, G. G. (2007). Simplified expressions for adjusting higher-order turbulent statistics obtained from open path gas analyzers. Boundary-Layer Meteorol., 122(1), 205–216.). Please also see the reply to **R2S21**, where we refer in more detail to the density correction applied.

We here use densities throughout the manuscript, these observations were collected with open-path analyzers. Since we focus on investigating the theoretical aspects of the REA approaches, we leave the physically correct conversion of densities into mixing ratios to the informed user/reader. Hence, our results are not affected by this important distinction as we stay with densities throughout.

We are adding the following sentence to the part of the Introduction where we introduce the REA method:

*"Note that the term "concentration"' refers to densities (expressed in e.g. mmol $m^{-3}$) throughout this paper."*

**R2S4)** P. 4, equation (4): Δw is not defined.

Thanks for spotting this! We added the following explanation:

*"$\Delta \overline{w}$ is the difference of the mean vertical wind while sampling into the up- and downdraft reservoirs, respectively."*

**R2S5)** Section 2.4: A discussion about the symmetrical vs. asymmetrical deadbands is missing.

Asymmetrical deadbands are only relevant in case of non-Gaussian flow and concentration statistics, which can be gleaned from investigating skewness and kurtosis as the 3rd and 4th central statistical moments. We here include an in-depth discussion of the kurtosis.

We are considering this comment and are addressing this in point in Section 2.4 as follows:

*"Here, we only consider symmetrical deadbands, presuming symmetrically distributed flow and concentration statistics. Effects of non-Gaussian distributed w' and s' can be gleaned from investigating higher central statistical moments."*

**R2S6)** P. 5, line 19: "applying a linear deadband to w'": please explain what is meant by "linear".

You are right, the term 'linear deadband' has not been introduced thoroughly in our text. Thank you for pointing us to this shortcoming. With a linear deadband, we mean a deadband linearly scaling with $\sigma_w$ (fixed fraction 'a', linear equation: deadband size = a*$\sigma_w$ + 0, where b=0 is the intercept).

We are including a more in-depth explanation:

*"When applying a linear deadband to w' (left panel in Fig. 1), no sample is taken if the magnitude of w' is below a certain threshold. This threshold can be held constant or adjusted dynamically in time. Dynamical adjustments are often done by scaling with the standard deviation of the vertical wind $\sigma_w$. The linear deadband appears as two horizontal lines in the quadrant plot in Fig. 1 (left panel), defined by the linear equation*

$$a \cdot \sigma_w + 0$$

*where a is a constant."*

**R2S7)** P. 5, line 22: "the deadband being proportional to the integral strength of the turbulent diffusive process". This is unclear. What is the "integral strength"?

The vertical velocity variance mathematically is the integral of the w-power spectrum. This nomenclature is common in micrometeorology (e.g. compare integral turbulence characteristics: $\sigma_w$/ u*, or integral turbulence intensity: $\sigma_w$/U).

**R2S8)** P. 5, lines 26-27: "Hyperbolic deadbands aim to exclude eddies with little flux contribution and maximize the concentration difference between the two sampling reservoirs.". This is not specific to hyperbolic deadbands since the same can be said regarding constant deadbands.

Agreed, linear deadbands have the same goal, however, for hyperbolic deadbands the above statement is even more applicable: There is a distinct mathematical difference. Hyperbolic deadbands filter in the w'c' plane (i.e. in the plane of instantaneous flux contribution given by the instantaneous cross-product), while constant deadbands filter only in the w' (or $\sigma_w$) space. In case w' is large and c' very small, then the flux contribution is small, and the sample may be discarded by the hyperbolic, but considered by the linear approach.

We suggest to rephrase the sentence as follows:

*"Hyperbolic deadbands are specifically designed to exclude eddies with little flux contribution and maximize the concentration difference between the two sampling reservoirs."*

**R2S9)** Figure 1: I suggest to plot the symbols with a single colour, and add lines of different colours to indicate the thresholds, rather than plotting couloured symbols. As it stands the Figure is ambiguous: when we look at, for example, the light blue 0.5 $\sigma_w$ threshold, all the red and brown dots should also be included in this class.

Thanks for the hint. We changed the figure accordingly:

[Figure]

*The adjusted Fig. 1: Schematic quadrant plots to visualize the application of linear (left) and hyperbolic (right) deadbands. Different colors show which data points are included for different deadband sizes. The white dot marks the origin in both panels. In the right-hand panel, solid red dots mark the mean w'/ $\sigma_w$ and mean $T_s$'/ $\sigma_T$ for up- and downdrafts when a hyperbolic deadband with H = 1.2 is applied. The white dashed lines in the right-hand panel connect the red dots with the coordinate system origin. The deviation from 180° of the angle spanned between these lines is a measure for the asymmetry of the sample distribution.*

**R2S10)** Figure 1: What represent the black lines?

The dashed black lines represent the coordinate system axes through the origin; The continuous black line (not present in the updated version) represented the line through the two dots  [mean(Ts' (w' < 0));

mean(w'(w' < 0))] and [mean(Ts' (w'>0)); mean(w'(w' > 0))], respectively, computed over all samples (no deadband applied). We decided to remove it from the updated version of Figure 1 because it did not add much value to this conceptual plot.

**R2S11)** Equation (1): Defined in that way, H is simply the correlation coefficient between w and s. The correct expression is to remove the overbar in (1) and mention that only the samples for which abs(H) is higher than a given threshold are retained.

You are right, we are changing the equation in question so that it matches Eq. (2) in Bowling et al. (1999):

$$H = \left| \left( \frac{w'}{\sigma_w} \right) \left( \frac{p'}{\sigma_p} \right) \right|$$

Written in this form, the relation to the hyperbolic graphs in the right panel of Fig. 1 also becomes more obvious. Please note that we have changed "*s*" to "*p*", to address the issue raised in **R2S12**.

**R2S12)** Equation (1) and throughout the text: "c" is used for an unspecified variable, but later on it represents the $CO_2$ concentration. This is confusing.

Thanks for the hint, which will certainly improve the comprehensibility of our manuscript. We decided to use "c" for $CO_2$, "s" for the scalar of interest, and "p" for the proxy scalar throughout the paper.

**R2S13)** Page 6, line 5: what is "n"?

n is the number of valid samples. We suggest to rephrase the part of the text in question as follows:

*"The use of large deadbands must be done with caution because they exclude a significant fraction of the data from being sampled. As a result, the random sampling error, which is related to $1/\sqrt{n}$, can be increased due to the decreased sample size n."*

**R2S14)** Page 7, last line of 2.4: I am not convinced that the angle reflects the asymmetry of the sample distribution. A highly-skewed sample distribution would be represented by dots at a really different distance from the (0,0) coordinate origin, but these dots could be aligned with the origin.

We agree that the deviation from the straight (180°) line (which we are using in our analysis) is not sufficient to describe the asymmetry of the sample distribution. We had a deeper look into our data, and computed an additional asymmetry measure as suggested in the comment. Below figure displays the difference in distance to the origin; i.e. we compute the distances of the red dots to the coordinate origin (0,0; white dot) in Fig. 1, and then look at the difference of these distances, which represents skewness of the sample distribution.

[Figure]

*Asymmetry measure (difference in distance of the two connecting lines to the origin) as a function of deadband size for linear deadbands (left) and hyperbolic deadbands (right).*

A similar pattern as visible in the two bottom panels of Fig. 4 emerges: The asymmetry increases significantly with increasing deadband size. One difference to Fig. 4 is that the values for the gravel site lie in the same range as for the other two sites, which was not true for the asymmetry expressed by the deviation from 180°.

However, as this evaluation does not add any new insights into the asymmetry dependence on deadband/ sample size, we propose to not include this figure in the paper. Instead, we suggest to simply rephrase the sentence introducing the asymmetry measure:

*"The asymmetry is shown as a white dashed line in the right panel of Fig. 1 containing a bend. This bend, which can be expressed as an angle deviating from 180°, is **one** measure for the asymmetry of the sample distribution."*

**R2S15)** Table 1: I suppose there is a mistake in the unit of roughness length.

Thank you for spotting this error! Indeed there was an error in our code to compute $z_0$ according to Eq. (2) in Panofsky (1984). The values are now 0.18 m, 0.06 m, and 4.87 m for the grassland, the loose gravel, and the forest site, respectively. We changed the numbers in Table 1 accordingly.

**R2S16)** Fig. 2: Replace "sensible heat flux" with "kinematic heat flux" and "latent heat flux" with "moisture flux"; or convert units into $Wm^{-2}$.

Figure 2 has been updated accordingly, thanks for the hint:

[Figure]

*The updated Fig. 2*

**R2S17)** Page 9: "During calm conditions, the region is dominated by a strong near-surface temperature inversion.". This is surprising, because Fig. 2 shows a positive (upward) heat flux, which is in conflict with a near-surface temperature inversion.

We are addressing this comment together with comment **R2S18** because they refer to the same issue.

**R2S18)** Page 9: "Strong katabatic winds draining the polar plateau frequently disrupt this inversion.". This is not consistent with the wind values given in Table 1.

We are addressing this comment together with the previous comment (**R2S17**). Both mentioned text passages refer to polar night conditions in Antarctica, which were not encountered during the field campaign used in our study. We removed the sentences from the manuscript.

We want to thank the reviewer for noticing this inconsistency. It is correct that during our campaign, which took place in Austral summer, we only encountered upward directed heat fluxes and no surface inversion at all.

**R2S19)** Page 10, line 13: Please explain what is a "dynamically determined lag.".

The sentence "Covariances were maximized by shifting the scalar time series relative to that of the vertical velocity by a dynamically determined lag.", refers to a step during data processing, where the time series of scalar concentration is shifted to achieve maximum cross-correlation with the vertical wind time series. This shift is determined dynamically, i.e. individually for each sampling period. After shifting the time series, the initial covariances can be calculated (see Foken, 2008, Micrometeorology, p. 109).

We are adding the following sentence to make this clearer in our text:

*"This means that, for each sampling period, the scalar time series were shifted to achieve maximum cross-correlation with the vertical wind time series (Foken, 2008)."*

**R2S20)** Page 10, line 20: Please explain what is meant by "additional hard thresholding was applied.".

We applied physical plausibility thresholds to filter the data for unphysical outliers. These thresholds were different for each scalar and each data set, due to different biochemical and meteorological conditions, and different measurement systems used. More specifically, the thresholds were defined as follows in our code (for the Dry Valleys/ gravel site "DRYVEXA", the forest site "WS2016", and the meadow site "ExpMM2015", respectively):

```
if (d=="DRYVEXA"){phys=c(0,400,0,300,-0.020,0.010)} # sensible heat, latent heat, co2

if (d=="WS2016"){phys=c(-100, 550,-100,400,-0.010,0.010)}

if (d=="ExpMM2015"){phys=c(-100, 200,-80,250,-0.020,0.010)}
```

To improve the readability of our text, we added an explanatory sentence to Section 3.2, which combines this issue with another comment raised by Referee #1 (**R1C4**):

*"In the final step, the same thresholds for physical plausibility which were applied to the computed EC fluxes were also used to remove unplausible REA flux estimates from the data sets. These thresholds were chosen individually for each scalar and each data set due to the wide range of meteorological and biochemical conditions covered in this study."*

**R2S21)** Page 10, lines 25-27 (also in line with comment#4 above): "To this end, molar densities were multiplied by the ratio of the instantaneous to mean density of moist air q <q>−1. EC fluxes were computed using the common post-hoc density correction (Webb et al., 1980).". This is unclear. Please check carefully. When one computes eddy-correlation fluxes with the scalar fluctuation expressed in mixing ratio unit (or any other proportional unit), the so-called WPL correction is not to be done. What I understand here is that the authors start with a mixing ratio, convert it to a quantity proportional to a density, and eventually apply the WPL correction (one step back, one step forward...).

Thank you for bringing up this point. It is always beneficial to review the data processing steps and check for consistency.

We start with densities (in mmol m$^{-3}$), which are being corrected by an *ad-hoc* density correction. *Ad-hoc* means that the correction (Detto and Katul, 2007) is directly applied to the high-frequency time series.

Starting from Eq. (3) in Detto and Katul (2007):

$$\rho'_{c,ext} = \frac{n_c}{n_a}\rho'_a$$

with

$$\frac{n_c}{n_a} = \frac{\overline{\rho_c}}{\overline{\rho_a}}$$

where overbars indicate means and primed quantities are fluctuations; $\rho_c$ is the density of a scalar c, $\rho_a$ the density of dry air; $\rho'_{c, ext}$ are the fluctuations of scalar c due to fluctuations in external conditions (mainly due to changes in air temperature and water vapor density). $n_c$ is the number of molecules of scalar c.

Combining the above equations and solving for the "correct" density, $\rho_{c} - \rho'_{c, ext}$ (which does not contain the fluctuations of external conditions), leads to the correction mentioned in our text, e.g. for $CO_2$:

$$CO_{2\,corr} = \frac{\overline{\rho_a}}{\rho_a} \cdot CO_2$$

This correction leaves the density units untouched (the ratio of fluctuating to mean pressure is dimensionless).

This *ad-hoc* correction is an alternative to the *post-hoc* WPL correction. However, since we sample in the instantaneous w's' planes, we need to account for the contraction-expansion argument (Detto and Katul, 2007) for every single sample. Our EC fluxes (to compute the $F_{REA}/F_{EC}$) ratios in turn were corrected using the default *post-hoc* WPL correction.

In summary, the first part of the quoted text in the comment refers to the *ad-hoc* density correction, which is needed for the REA simulator. The WPL correction was only applied for computation of the EC fluxes. We agree that this passage reads somewhat confusing and have tried to rephrase it in a more understandable way as follows:

*"Since simulating REA sampling requires selecting individual high frequency data from a continuous time series and computing density-corrected scalar higher-order moments, an ad-hoc density correction was applied to the water vapor and carbon dioxide molar densities (Detto and Katul, 2007) prior to flux computations. To this end, molar densities were multiplied by the ratio of the instantaneous to mean density of dry air* $\rho_a \overline{\rho_a}^{-1}$ *. This correction removes the density fluctuations due to changes in external conditions. EC fluxes were computed using the common post-hoc density correction (Webb et al., 1980)."*

**R2S22)** Figure 4: "valid samples" is not the most appropriate terminology. Rather "selected samples", or something equivalent…

Agreed, we changed the figure accordingly.

**R2S23)** Figure 5: Why a logarithmic scale on the right panel ? RMSE unit is missing.

Due to the large variety of meteorological conditions covered by the three data sets, we obtain a large range of RMSE values. We felt that a logarithmic scale makes the plot clearer than a linear scale. However, after more thorough data cleaning in response to a comment by referee #1, the ranges covered by the RMSE are not that large anymore. We are changing the y axes of Figs. 5-8 to linear scale, see the updated Fig. 5 below. Thank you also for spotting the missing unit. We updated the figure accordingly:

**Dynamic linear deadband with $\beta_T$ (model 1)**

[Figure]

*The updated Fig. 5. The missing unit of the RMSE was added and the y axis scale in the right panel was changed to linear.*

**R2S24)** Figure 10: It is unclear how the stability bins are defined. It seems that the neutral class is very large, encompassing stable and unstable conditions until abs(z/L)~0.1. There is therefore an evidence of overlapping between the classes.

Originally, we wanted to define a larger "neutral" class (with intended overlapping), however we see that this does not really provide any additional insight. We reduced the neutral stability class to z/L values between -0.0177 to + 0.0177 (or +/- $10^{-1.75}$), and removed the vertical dashed lines from Fig. 10:

[Figure]

The bins are defined as logarithmically evenly spaced classes of dynamic stability. We split the range of encountered values of dynamic stability as shown in the below schematic. By defining the boundaries as $-10^{0.25}$, $-10^{-0.25}$, $-10^{-0.75}$, … the centers of the bins (on a logarithmic axis) are the red values, which correspond to the labels on the x-axis of Fig. 10. The numbers in the lower part of the schematic indicate the number of samples in each bin:

[Figure]

We suggest to include the schematic as a table in the appendix (Appendix B). We are adding the following explanation to our text to improve the clarity:

*"These classes were defined such that the range of dynamic stability spanned by each bin is equally sized in the logarithmic space."*

**R2S25)** Figure 10, caption: "bars" instead of "arrows".

We exchanged the "arrows" with "bars" in the figure caption, and also replaced all the other occurrences in the text.

**R2S26)** Figure 11: It is not clear which β model is represented here. Explain in the caption what is the grey zone.

Thanks for the comment. We are changing the caption as follows:

*"This figure only presents results from REA model 3 ($\beta_w$). Left panel: $\beta_w$ as a function of w' kurtosis for different deadband widths (not binned). Valid data points from all three sites are combined in this panel. Center panel: the stability parameter z/L as a function of the w' kurtosis. Data were binned into eight kurtosis bins with equivalent number of data points. Only bin medians are displayed, bars mark the IQR. Right panel: Median $F_{REA}/F_{EC}$ as a function of w' kurtosis for the optimal deadband widths, 0.9 $\sigma_w$ and 0.5 $\sigma_w$, which were determined by Baker (2000) and in this study. Data were grouped into the same kurtosis bins as in the center panel. The grey area marks the +/- 10% range, which is the error assumed in EC applications."*

**R2S27)** Page 20: "increasing z/L" is ambiguous since z/L could be either positive or negative

We are changing the wording to:

*"with increasing (positive) z/L"*

**R2S28)** Page 2; line 9: replace "sensible heat" and "latent heat" with "temperature" and "water vapour", respectively.

Thanks, we took this comment into account and changed the wording accordingly.

**R2S29)** Page 21: "methods. The diurnal course of the flux bias showed large deviations from the EC flux, particularly during transitions when the direction of the flux changed". This is unclear, please rephrase.

Thanks for the hint. We are rephrasing the sentence as follows:

*"However, during times of low proxy-scalar correlation, the variability of this ratio, measured by the RMSE, was large. This happened particularly at those times of the day when the direction (sign) of the flux changed."*

**R2S30)** Conclusion: A common name should be used for each model between the text and in Table 2.

We want to thank the reviewer for this comment, which will certainly improve the comprehensibility of the conclusions section. We are adding references to the model numbers 1-4, which are listed in Table 2, to the text.

**R2S31)** Section "Conclusions and practical recommendations": The last sentence of the abstract contains a recommendation which is not present here.

Thank you for the hint - this recommendation should indeed be pointed out more directly! We propose to add the following to the Conclusions, and thus conclude the paper as follows:

*"Based on the findings obtained in this study, we attempt to formulate the following general recommendations: For applications without deeper site-specific knowledge, we recommend using either the $\beta_w$ or $\beta_{T, const}$ approach (model 3 or model 4). These two models have been shown to perform robustly and be less sensitive to changes in proxy-scalar similarity than model 1 and 2. In case of a well-known site, including scalar-scalar similarity, we propose to use the proxy-dependent approach in connection with a hyperbolic deadband (model 2). Model 2 yielded very similar results to model 1 with respect to the precision and accuracy measures considered in this study. However, hyperbolic deadbands are better suited to maximize the concentration difference between up- and downdraft reservoirs, which is of advantage when investigating fluxes of compounds with very low atmospheric concentrations."*

---

## Referee Report (RR1)

The authors addressed the major comments of my first review in a more or less satisfying way (see detailed comments below). However, they overlooked or disregarded all my "Minor Comments", which is not acceptable. In addition, the author responses (and revisions) led to some additional issues that need to be addressed before the paper can be published.

*Important*: The page and line numbers I use in the following refer to the revised manuscript version with markups (ATC1).

COMMENTS

1) Please address the list of "Minor comments" in my first review. These comments have not been considered in this revised version (Some of the comments may have become obsolete due to other changes)

2) R1C1 response: I still miss the important discussion (in the Discussion section) about the assumption of scalar similarity, i.e. whether and why the results for H2O investigated here (with minor results also for CO2) can be applied to all other scalars, especially the ones for which REA is usually applied.
This is especially critical because the authors state on P24, Line 8: "Choosing the optimum proxy scalar is critical for the methods success". This sentence implies that a general similarity between all scalars is not expected.

3) P1, line9: In the context of major comment 2 above, this statement is over-ambitious especially concerning the part "...formulating universally applicable recommendations...", and I suggest to downgrade it to some extent.

4) R1C4 response: Such a strong effect of one single outlier makes the suitability of the used evaluation method questionable.

5) R1C6 response: The pooling of data from all three sites in the evaluation can be problematic. Does this imply that $\beta_w$ and the w-statistics as well as $\beta_T$ values are fully independent of the site conditions (canopy height, roughness, correlation between proxy scalar and scalar of interest, etc.) ? This issue needs some statements/discussion in the text.

6) R1C5 response: The newly added Table 2 is very important for the present study. There are two important questions arising from it that deserve some thoughts/discussion: i) can the average $b_w$ be considered as a site independent constant?;  ii) Would the use of an average constant $b_w$ yield similarly good results like the use of half-hourly $b_w$-values?
The (successful) use of an overall constant $b_w$ value would strongly simplify the REA measurements.

7) In various parts of the text and the abstract the terms "proxy-based approaches" or "proxy-dependent approaches" are used synonymously for the models 1 and 2. However, this is misleading and not correct, because the model 4 approach is proxy-based as well (and also model 3 uses w as proxy, though it is not a scalar). I suggest to use a more suitable and specific expression for models 1 and 2 like e.g. "dynamic scalar proxy approaches" or just "model 1 and 2 approaches"

8) Figure 10: correct the y-axis titles in the left and central panel to $\beta_T$.

9) Figure A1: For H2O in Fig. 9 an individually adjusted deadband width was used. Was the same deadband width also applied for CO2, or was it indivdually adjusted for CO2. In either case, does an individual adjustment yield the same optimum deadband widths for H2O and CO2?

---

## Author Response (AR2)

**Reply to comments of Reviewer # 1:**

We want to thank the anonymous referee again for their helpful comments. We indeed have missed to address the minor comments from their first review, which was due to human error, which we want to apologize for. We are addressing the raised comments (and the minor comments from round 1) in a point-by-point way below:

*The authors addressed the major comments of my first review in a more or less satisfying way (see detailed comments below). However, they overlooked or disregarded all my "Minor Comments", which is not acceptable. In addition, the author responses (and revisions) led to some additional issues that need to be addressed before the paper can be published.*

*Important: The page and line numbers I use in the following refer to the revised manuscript version with markups (ATC1).*

*Again, we apologize for missing out on addressing the minor comments. We are addressing them in this review round.*

*COMMENTS*

*1) Please address the list of "Minor comments" in my first review. These comments have not been considered in this revised version (Some of the comments may have become obsolete due to other changes)*

*We are addressing the minor comments from round 1 in a point-by-point way below:*

*MINOR COMMENTS*

*P1, L15-16: It is not clear, which β approach this sentence is related to.*

*This comment has become obsolete. We agree that this sentence was unclear, and it was removed from the abstract.*

*P2, L2-5: Both sentences are formulated in a misleading way.*

*First sentence: the detection limit of the instrument does not limit the REA fluxes directly but the quality/uncertainty of the REA fluxes. Change e.g. to "...when the uncertainty of the REA flux quantification is not limited by ...".*

*Second sentence: change to: "For REA sampling differences close to the instruments detection limit …"*

*Thank you for pointing us to these unclear formulations. We appreciate your suggesting a better alternative. We have changed the sentences accordingly.*

*P6, Fig. 1: Please check if the position of the grey points in the right panel is correct. According to Eq. 4 and a β w value of about 0.6, the normalized vertical distance of the two grey points (=DELTAw/ SIGMA w)  should be about 1.6, but in the figure this distance is much less than 1. Maybe the x- and y-axis need to be exchanged...?*

*Fig. 1 was updated during the first review round, and the grey points were removed. Please note also that the right panel in Fig. 1 refers to hyperbolic deadbands, and (former) Eq. 4 refers to the β w method (model 3).*

**P8, Table 1: The units for the roughness length are probably [cm], not [m]. Only indicate two significant digits in the roughness length values, because their accuracy is not so high.**

**Please also include the average canopy heights and the EC measurement heights in the table (better than scattered in the text). This would be advantageous for the reader.**

*Thank you for spotting the error and for the suggestion to include canopy and measurement heights in Table 1. The roughness length values were updated during the first review round (comment R2S15). We have added the estimated canopy and measurement heights in the table as suggested.*

**P9, L17: The formulation "...resulting in a total measurement height ..." is not logical (what results in what?). Please rephrase.**

*We have rephrased the sentence as follows: „The EC flux instrumentation (2 m high) was installed on top of a 31 m high scaffolding tower reaching above the highest tree tops, resulting in a total measurement height of 33 m above ground.“*

**P9, last line: Correct to: "A diel course is still observed, but the flux is constantly directed ..."**

*Thanks for spotting this. We have changed the sentence as suggested.*

**P10, L18: What do you mean with "perturbation time scale"**

*„perturbation time scale“ is a term related to Reynolds averaging. It describes the **time scale** over which the average is computed; this average is used to derive the **perturbations** by subtracting it from each individual observation.*

**P10, L25: Explain the "additional hard thresholding".**

*This issue was addressed in the answer to R2S20 as follows:*

We applied physical plausibility thresholds to filter the data for unphysical outliers. These thresholds were different for each scalar and each data set, due to different biochemical and meteorological conditions, and different measurement systems used. More specifically, the thresholds were defined as follows in our code (for the Dry Valleys/ gravel site "DRYVEXA", the forest site "WS2016", and the meadow site "ExpMM2015", respectively):

```
if (d=="DRYVEXA"){phys=c(0,400,0,300,-0.020,0.010)} # sensible heat, latent heat,
co2

if (d=="WS2016"){phys=c(-100, 550,-100,400,-0.010,0.010)}

if (d=="ExpMM2015"){phys=c(-100, 200,-80,250,-0.020,0.010)}
```

To improve the readability of our text, we added an explanatory sentence to Section 3.2, which combines this issue with another comment raised by Referee #1 (**R1C4**):

*"In the final step, the same thresholds for physical plausibility which were applied to the computed EC fluxes were also used to remove unplausible REA flux estimates from the data sets. These thresholds were chosen individually for each scalar and each data set due to the wide range of meteorological and biochemical conditions covered in this study."*

**P11, Line 1-2: I do not understand what "force it through zero" means here.**

*This question is related to the problems arising from the negligible $CO_2$ flux measured at the Antarctic gravel site. We added a constant correction of $0.00035$ mol $L^{-1}$ to each observation to obtain a mean $CO_2$ flux of 0. This value was determined empirically. Using this constant offset correction does not impact our other evaluations and was only done for physically correct visual representation, as no $CO_2$ flux is expected at that site due to absence of biological activity.*

**P11, Line 5: I do not understand why the slope m had to be computed in the present study. It is not necessary for the β w calculation according to Eq. 4. Moreover, the w'-c' statistics are not available in a real REA application (see also comment 3 above).**

*You are correct that m does not need to be computed, and that w'-c' statistics are not available for real REA applications. We included these considerations for the sake of completeness, and for clarity of the theoretical derivation of REA / β.*

**P20 Fig. 11 middle and right panel: The symbol colors hardly distinguishable. Removing the black frame of the symbols may be helpful.**

*Thank you for the suggestion. We decreased the line width of the markers used in this plot.*

**P21, L7-10 ("The tested REA models .... along with the main results of this study.") This part should be omitted from the Conclusions because it is pure repetition.**

*We want to thank the reviewer for this comment. However, we think that this summary of the used methods and models at the beginning of the Conclusions is beneficial for the ‚quick reader' focusing on abstract and conclusions, and hence would rather like to keep this part. We leave the decision to the editor.*

**Figures 5-8: Indicate the units of the RMSE in the right panels.**

*Thank you for spotting the missing unit in the plots. They were added to the figures.*

**Table 2: In the second lowest row, "rxx" presumably should be replaced by "rxy"**

*Thank you for spotting this. The term „$r_{xy}$" was changed to „$r_{sp}$" (scalar – proxy)  during the first review round, including in the table (where the phrase was wrong indeed)*

*2)*

*The reviewer's second comment refers to the discussion following R1C1 during the first review round, which is why we are first adding the complete discussion (R1C1 and our answer to this comment) below:*

**round 1 R1C1)** Only the performance of the REA approaches for the $H_2O$ flux is tested in the present study. This is done after an initial deadband optimization (using the reference EC dataset) for the same test scalar. This leads to a certain lack of independence in the method validation. Although the $CO_2$ flux and its correlation with the other scalar fluxes is introduced in Sections 3 and 4.1, the REA evaluations for the $CO_2$ flux are unfortunately not presented. Alternatively $CO_2$ could have served as second proxy scalar option beside the temperature T (at least for some sites) as indicated in Section 2.3.

The authors should more prominently (in abstract and objectives) declare that they are evaluating the REA approaches only for $H_2O$ fluxes. In addition they need to discuss better, whether and why they assume that the results also apply to other scalars, despite a sometimes low scalar correlation as exhibited in Fig. 3.

**round 1 authors' answer to R1C1)** The reason why only the results for the $H_2O$ flux are presented was to limit the analysis to a reasonable scope. Additionally, we decided to not present the $CO_2$ flux results because, for the gravel site (Antarctica), there is basically no measurable $CO_2$ flux due to lack of biological activity, which makes the interpretation difficult. However, we agree that, for method validation, considering another flux than the one for which the deadband size was optimized is required. Following the referee's suggestion, we propose adding an appendix (Appendix A), in which we present the hourly binned RMSE evaluation, which was done for $H_2O$ in Fig. 9, but for the $CO_2$ flux. Alternatively, the below figure and interpretation could be included and discussed in the main manuscript. We would like to leave this decision to the editor. Regarding the second part of the comment, we state that the changes will be reflected in abstract and introduction.

[Figure]

*Fig. 11: Same as Fig. 9 but for the CO$_2$ flux. The gravel site results (solid black lines) should be regarded with caution as the magnitude of the CO$_2$ flux at this site is close to zero (compare to Fig. 2).*

Interpretation: *The same findings that were drawn from the H$_2$O flux analysis are also apparent in the above figure: Both proxy approaches (panels (a) and (b) ) result in higher values of the RMSE than the $\beta_w$ (panel (c) ) and the constant $\beta$ (panel (d) ) methods. The RMSE for both proxy approaches at the meadow site peaks during 13-14 UTC, the time when scalar-scalar correlation of sensible heat and CO$_2$ is lowest. At the forest site, the RMSE for the $\beta_T$ approaches is highest when the magnitude of the CO$_2$ is largest. The RMSE for the gravel site is included in this figure even though the magnitude of the CO$_2$ flux is close to 0 throughout the daily course and thus no conclusions should be drawn from its RMSE.*

**R1C1 response: I still miss the important discussion (in the Discussion section) about the assumption of scalar similarity, i.e. whether and why the results for H$_2$O investigated here (with minor results also for CO$_2$) can be applied to all other scalars, especially the ones for which REA is usually applied.**

**This is especially critical because the authors state on P24, Line 8: "Choosing the optimum proxy scalar is critical for the methods success". This sentence implies that a general similarity between all scalars is not expected.**

Answer to the answer to R1C1:

We want to thank the reviewer for this comment. First of all, we would like to clarify that two different „instances" of scalar similarity are of importance in the presented study:

(i) scalar-proxy similarity assumed by models 1 and 2, which employ a half-hourly adjusted proxy-derived β p value according to Eq. (2):

$$\beta_p = \frac{\overline{w'p'}}{\sigma_w \cdot \Delta\overline{p}}$$

We can investigate the validity of this scalar-proxy similarity assumption e.g. using the scalar-proxy correlation coefficients $r_{s,p}$

(ii) scalar similarity with respect to the general validity of our presented results for scalars different from $H_2O$. E.g. whether the optimized deadbands presented in this study can be used also for REA flux measurements of other atmospheric compounds such as ammonia or aerosol particles. This is more difficult to answer, regarding the available data we have.

You are encouraging more in-depth discussion about (ii), while the sentence you cite from the Conclusions refers to (i). We are therefore suggesting to rephrase the sentence as follows:

„*Concerning models 1, and 2, choosing the optimal proxy scalar is critical for the methods' success.*"

It is however true that we did not further discuss (in the Discussion section) whether conclusions can be drawn about the flux estimation of other scalars than $H_2O$. We acknowledge that adding a more in-depth discussion about this issue would definitely improve the manuscript. Given the data we have at hand (fast-response observations of $CO_2$, $H_2O$ and temperature), we can however only answer the question whether our results found for the $H_2O$ flux are also valid for the $CO_2$ flux, if temperatures is used as the proxy .

We have recreated Figs. 5-8 but for the $CO_2$ flux:

Dynamic linear deadband with $\beta_T$ (model 1)

[Figure]

*Figure 1: Errors as a function of dynamic linear deadband width. The x axis is the scaling factor a multiplied with the vertical wind standard deviation in Eq. 9 to define the deadband threshold. Left panel: Median $F_{REA}/F_{EC}$ ($CO_2$ flux simulated with sensible heat as a proxy) ratio for each of the simulated dynamic deadband widths; right panel: RMSE for each of the simulated dynamic deadband widths*

Dynamic hyperbolic deadband with $\beta_T$ (model 2)

[Figure]

*Figure 2: Errors as a function of dynamic hyperbolic deadband size. The x axis is the H parameter in Eq. 10, which defines the deadband size. Left panel: Median $F_{REA}/F_{EC}$ ($CO_2$ flux simulated with sensible heat as a proxy) ratio for each of the simulated dynamic deadband sizes; right panel: RMSE for each of the simulated dynamic deadband sizes*

Dynamic linear deadband with $\beta_w$ (model 3)

[Figure]

Figure 3: Errors as a function of dynamic linear deadband width. The x axis is the scaling factor a which is multiplied with the vertical wind standard deviation in Eq. 9 to define the deadband threshold. Left panel: Median $F_{REA}/F_{EC}$ ($CO_2$ flux simulated using the REA approach described in Baker (2000)) for each of the simulated dynamic deadband widths; right panel: RMSE for each of the simulated dynamic deadband widths

Dynamic linear deadband with constant $\beta_T$ (model 4)

[Figure]

Figure 4: Errors as a function of dynamic linear deadband width. The x axis is the scaling factor a which is multiplied with the vertical wind standard deviation in Eq. 9 to define the deadband threshold. Left panel: Median $F_{REA}/F_{EC}$ ($CO_2$ flux simulated using constant $\beta$ T and dynamic linear vertical wind deadband) for each of the simulated dynamic deadband widths; right panel: RMSE for each of the simulated dynamic deadband widths

The optimal deadband sizes summarized in Table 2 prove to be also valid for the $CO_2$ flux: While it is hard to decide for an optimum deadband size for models 1 and 2, the deadband sizes found for $H_2O$ (a=0.5; H=0.5) are among the best-performing options. For models 3 and 4, the deadband size found for $H_2O$ (a=0.5 for both models) also exhibit an optimum (minimum) RMSE for the $CO_2$ flux.

During the first review round, we showed that applying these deadband sizes to $CO_2$ results in a similar pattern in the diurnal RMSE as was observed for the $H_2O$ flux: Smaller RMSE for models 3 and 4 than for models 1 and 2, and the forest site exhibiting larger RMSE than the meadow site.

We propose adjusting the paragraph about the $CO_2$ flux at the end of section 4.3.1 as follows, adding a discussion about the applicability to other scalars:

*„So far, only one proxy-scalar combination was investigated in this study. However, showing that the presented results are also valid for other scalars is critical for their applicability. The data sets allow for including $CO_2$ for additional validation. The $CO_2$ flux was simulated with the optimized models 1-4 (using the deadband sizes summarized in Table 2), with sensible heat as the proxy for models 1, 2 and 4. Comparison of $F_{REA}/F_{EC}$ ratio and RMSE indicated that the optimum deadband sizes found for $H_2O$ (Table 2) are also valid for $CO_2$. The hourly RMSEs are included in Appendix A in Fig. A1. A similar pattern in the diurnal RMSE as observed for the $H_2O$ flux also emerges for $CO_2$: Models 1 and 2 both yield higher RMSEs than models 3 and 4, the forest site exhibiting larger RMSEs than the meadow site. These findings suggest that the results presented here for the $H_2O$ flux are also valid for the $CO_2$ flux, and possibly other atmospheric compounds. However, we cannot arrive at a final conclusion for other (including reactive) scalars, for which REA is often applied, since fast-response analyzers are missing. Answering this question is beyond the scope of this study and should be considered in future research.“*

**3) P1, line9: In the context of major comment 2 above, this statement is over-ambitious, especially concerning the part "...formulating universally applicable recommendations...", and I suggest to downgrade it to some extent.**

We want to thank the reviewer for this comment. We propose to change the sentence as follows:

*„This study evaluates a variety of different REA approaches with the goal of formulating recommendations applicable over a wide range of surfaces and meteorological conditions for an optimal choice of the β factor in combination with a suitable deadband.“*

**4)**

*The fourth comment refers to the discussion following R1C4 during the first review round, which is why we are first adding the discussion regarding R1C4:*

**R1C4)** How can it be that the zero deadband calculations result in RMSE of about 20 mmol m-2 s-1 for the forest site in Figs. 5 and 6, when the fluxes themselves are only between 0 and 4 mmol m-2 s-1 (Fig. 2) and the flux ratios in the left panels are close to 1? This seems very unplausible and needs a detailed explanation.

**round 1 authors' answer to R1C4)** Thanks for spotting this. The large RMSE compared to

the median $F_{REA}/F_{EC}$ ratio close to 1 was actually due to one single outlier. We decided to take the physical plausibility thresholds, which were applied to the EC fluxes, and also apply them to all simulated REA fluxes. This removes the outlier in question, and reduces the RMSE values for the forest site in Figs 5 and 6. However, the thresholding does not alter any of the other presented results significantly. The main finding presented in this section, i.e. that the proxy-based approaches result in a larger error compared to the $\beta_w$ and $\beta_{T,const}$ approaches, remains still valid.

We propose to include the following explanation in Section 3.2, stating that the physical plausibility thresholds were applied to the simulated REA fluxes as well:

*"In the final step, the same thresholds for physical plausibility which were applied to the computed EC fluxes were also used to remove unplausible REA flux estimates from the data sets. These thresholds were chosen individually for each scalar and each data set due to the wide range of meteorological and biochemical conditions covered in this study."*

**R1C4 response: Such a strong effect of one single outlier makes the suitability of the used evaluation method questionable.**

We agree that the RMSE is prone to outliers in the data, which makes strict qualtity criteria necessary. Due to concerns regarding RMSE, we decided to not only make our choice of optimal deadband using the RMSE but to also take median $F_{REA}/F_{EC}$ into account (which did not change after removing the outlier, but in turn are problematic when the magnitude of observed fluxes is small). Combining these two measures makes us confident with regards to the choice of optimal deadband size. The sensitivity of the RMSE to individual large outliers is inherent to its statistical definition, which is a well accepted metric when comparing methods.

**5)**

*The fifth comment refers to the discussion following R1C6 during the first review round, which is why we are first adding the discussion regarding R1C6:*

**R1C6)** For Figure 10 and 11 it is not indicated, which data are displayed. Are these all (valid) data for all three sites or only data from one site? This needs to be clearly stated in the Figure caption.

**round 1 authors' answer to R1C6)** Thanks for bringing up this issue. In Figs 10 and 11, all valid data from all three sites are combined. The observations from all three ecosystems fall along the same lines, which suggests that e.g. the findings of $\beta_w$ vs. kurtosis as a function of deadband size presented in the left panel of Figure 11 are universally applicable.

For clarification, we are adding the following sentence to the caption of Fig. 10:

*"This figure combines valid data points from all three sites."*

and we are adding

*"Valid data points from all three sites are combined in this panel."*,

to the caption of Fig. 11.

**R1C6 response: The pooling of data from all three sites in the evaluation can be**

*problematic. Does this imply that βw and the w-statistics as well as βT values are fully independent of the site conditions (canopy height, roughness, correlation between proxy scalar and scalar of interest, etc.) ? This issue needs some statements/discussion in the text.*

We want to thank the reviewer for this comment. This is a very interesting point indeed. The findings of Ammann & Meixner (2002) pointed towards a systematic dependence of the $\beta_p$ factor from the stability parameter z/L which we tried to reproduce in our data. However, we can only find the pattern described in their work for the forest site, and only if no or small deadbands are used. For the other sites and larger deadbands, the relationship described by Ammann & Meixner (2002) vanishes. Below are the plots of the results shown in Fig. 10, but for each site individually:

[Figure]

*Figure 5: Same as Figure 10, but for the forest site only*

[Figure]

*Figure 6: Same as Figure 10, but for the meadow site only*

[Figure]

*Figure 7: Same as Figure 10, but for the gravel site only*

However, concerning the $\beta_w$ factor, the situation is different. The right-hand panels in the above figures all show a strikingly similar pattern; also, if we are plotting the data in the left panel in Fig. 11 (which was the core of this comment) individually for the three sites, the results look very similar:

[Figure]

*Figure 8: The left panel of Fig. 11 but for each site individually: Forest (left), meadow (center) and gravel (right).*

To answer the reviewer's question: Yes, this implies that $\beta_w$ and the w-statistics are fully independent of the site conditions. $\beta_T$ on the other hand behaves differently for each site and cannot be described by z/L reliably.

We propose rephrasing part of the discussion about Fig. 10 as follows:

*„For the two dynamic proxy models (models 1 and 2; left and center panel in Fig. 10), $\beta_T$ without deadband approximately follows the relationship found by Ammann and Meixner (2002), i.e. a constant β T for unstable conditions, and an increase from neutral and stable conditions of z/L>= 0.06. However, this increase is associated with large statistical uncertainty and only due to the data from the forest site (please note that Fig. 10 combines the observations from all three sites). We therefore recommend exercising caution when using stability-dependent parameterizations of $\beta_T$. Variability of $\beta_T$ generally decreases with increasing deadband size. Model 3 (right panel in Fig. 10) shows a very different behavior: $\beta_w$ is apparently unrelated to dynamic stability, and displays a generally lower variabiliy than $\beta_T$.“*

Also, we propose to add the following to the discussion belonging to Fig. 11:

*„This finding suggests that the turbulence statistics, including the $\beta_w$ factor, are site-independent despite the significant differences in climate and surface characteristics across the three ecosystems (canopy height, roughness, etc.).“*

**6) R1C5 response: The newly added Table 2 is very important for the present study. There are two important questions arising from it that deserve some thoughts/discussion: i) can the average β w be considered as a site independent constant?; ii) Would the use of an average constant β w yield similarly good results like the use of half-hourly β w-values?**

**The (successful) use of an overall constant β w value would strongly simplify the REA measurements.**

Thank you for pointing this out. These are indeed very interesting questions. Concerning (i):

$\beta_w$ indeed was found to be independent from the site (see the answer to your comment above); and regarding (ii): It is shown e.g. in Fig. 11 that $\beta_w$ does not vary a lot over time.

We hence propose to add the following to the discussion of Fig. 11:

*„This is confirmed by the nearly identical average $\beta_w$ values found for the three sites in Table 2 of 0.43-0.44. In connection with the small spread of $\beta_w$ values in Fig. 11, and the strikingly similar RMSE for models 3 and 4 in Fig. 9, our results suggest that $\beta_w$ can be considered a both site- and time-independent constant.“*

**7) In various parts of the text and the abstract the terms "proxy-based approaches" or "proxy-dependent approaches" are used synonymously for the models 1 and 2. However, this is misleading and not correct, because the model 4 approach is proxy-based as well (and also model 3 uses w as proxy, though it is not a scalar). I suggest to use a more suitable and specific expression for models 1 and 2 like e.g. "dynamic scalar proxy approaches" or just "model 1 and 2 approaches"**

We thank the reviewer for this remark. However, we think that „dynamic scalar proxy approaches“ would impact the readability of our manuscript in a negative way. We have added the word „dynamic“ to the occurrences of „proxy-based“ and „proxy-dependent“, to distinguish these two models from model 4, which uses a constant proxy-based $\beta$ factor.

**8) Figure 10: correct the y-axis titles in the left and central panel to $\beta_T$.**

Thanks for spotting this. We have made the suggested changes.

**9) Figure A1: For $H_2O$ in Fig. 9 an individually adjusted deadband width was used. Was the same deadband width also applied for $CO_2$, or was it indivdually adjusted for $CO_2$. In either case, does an individual adjustment yield the same optimum deadband widths for $H_2O$ and $CO_2$?**

We want to thank the reviewer for this good comment. We have already answered this question in our reply to (1).